# Comments on abelian Higgs models and persistent order

Zohar Komargodski[1,2], Adar Sharon[1], Ryan Thorngren[3] and Xinan Zhou[4]

**1** Department of Particle Physics and Astrophysics,
Weizmann Institute of Science, Rehovot 7610001, Israel
**2** Simons Center for Geometry and Physics,
Stony Brook University, Stony Brook, NY 11794, USA
**3** Department of Mathematics, University of California,
Berkeley, California 94720, USA
**4** C. N. Yang Institute for Theoretical Physics,
Stony Brook University, Stony Brook, NY 11794, USA

## Abstract

A natural question about Quantum Field Theory is whether there is a deformation to a trivial gapped phase. If the underlying theory has an anomaly, then symmetric deformations can never lead to a trivial phase. We discuss such discrete anomalies in Abelian Higgs models in 1+1 and 2+1 dimensions. We emphasize the role of charge conjugation symmetry in these anomalies; for example, we obtain nontrivial constraints on the degrees of freedom that live on a domain wall in the VBS phase of the Abelian Higgs model in 2+1 dimensions. In addition, as a byproduct of our analysis, we show that in 1+1 dimensions the Abelian Higgs model is dual to the Ising model. We also study variations of the Abelian Higgs model in 1+1 and 2+1 dimensions where there is no dynamical particle of unit charge. These models have a center symmetry and additional discrete anomalies. In the absence of a dynamical unit charge particle, the Ising transition in the 1+1 dimensional Abelian Higgs model is removed. These models without a unit charge particle exhibit a remarkably persistent order: we prove that the system cannot be disordered by either quantum or thermal fluctuations. Equivalently, when these theories are studied on a circle, no matter how small or large the circle is, the ground state is non-trivial.

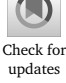
# 1  Introduction and Summary

Classical statistical systems can often be studied approximately using Quantum Field Theory (QFT). The larger the correlation length in lattice units, the better the approximation. Changing the temperature of the statistical model corresponds to deforming the QFT by a certain operator, usually one that respects all the symmetries.

The quintessential example of this correspondence between statistical systems and Quantum Field Theory is the Ising model in $d$ dimensions, whose equilibrium partition function is given by $Z = \sum e^{-\beta H}$ with $H = -J \sum_{\langle ij \rangle} s_i s_j$ defined on a $d$ dimensional hypercubic lattice, where $\sum_{\langle ij \rangle}$ is a sum over all the nearest neighbors and $s_i \in \{-1, 1\}$. The corresponding Quantum Field Theory is, loosely speaking, given by the action

$$S = \int d^d x \left( (\partial_\mu \sigma)^2 + m^2 \sigma^2 + \sigma^4 \right) .$$

If $m^2 > 0$ (in the sense of being positive and sufficiently large) then there is a unique vacuum with the $\mathbb{Z}_2$ symmetry $\sigma \to -\sigma$ being unbroken. If $m^2 < 0$ (in the sense of being negative and sufficiently large) then the $\mathbb{Z}_2$ symmetry is spontaneously broken. In the context of the statistical model, this corresponds to whether the temperature is higher or lower than the critical temperature, respectively. Equivalently, we could say that the temperature is associated with the relevant operator $\sigma^2$.

This general relationship between statistical models and QFT, the Ginzburg-Landau framework and its generalizations, combined with our intuition that at high temperatures in statistical systems all the symmetries of the Hamiltonian are obeyed, leads one to expect that QFT should generally have a phase for which the symmetries are restored.

This is however too naive. Some QFTs have 't Hooft anomalies for their global symmetries, and this precludes a ground state which is symmetric, gapped, and trivial [1]. We call these states "trivial" for short. For example, Yang-Mills theory with massless fermions has such an anomaly and this has been used to exclude a trivial ground state (see [2] and references therein). Another example is pure $SU(N)$ Yang-Mills theory at $\theta = \pi$ [3]. In this sense, such theories are outside the usual Ginzburg-Landau framework.

Theories without trivial phases play a very prominent role also in the context of deconfined criticality [4–11] in condensed matter physics. For some earlier related work see also [12,

13]. Here we study the simplest such examples and point out that the underlying mechanism behind the absence of a trivial phase is a 't Hooft anomaly. We also find that such theories, slightly modified (or in the presence of certain chemical potentials), have anomalies that are sufficiently powerful to preclude a trivial phase at finite temperature.

We discuss a class of such theories constructed from gauge fields coupled to scalar matter. To be precise, we use a $U(1)$ gauge field $a$ and $N$ charge $p$ complex scalars $\phi_i$, $i = 1, ..., N$. We will discuss this model in both 1+1 and 2+1 dimensions. The $p = 1$ models are most familiar, but $p > 1$ has some interesting features not seen at $p = 1$, i.e. the center symmetry. The Lagrangian in three dimensions is

$$\mathcal{L} = -\frac{1}{4e^2}|da|^2 + \sum_i |D_a \phi_i|^2 + \lambda \left( \sum_i |\phi_i|^2 \right)^2 , \qquad D_a \phi = \partial \phi - ipa\phi . \tag{1}$$

And in two dimensions we can also add a $\theta$ term

$$\delta \mathcal{L}_{2d} = \frac{\theta}{2\pi} \int da . \tag{2}$$

while this term is a total derivative, since $a$ is not a globally-defined 1-form when the gauge bundle is non-trivial, it can actually contribute to the non-perturbative physics. Because the integral of the gauge curvature $da$ is a $2\pi$-integer on any closed surface, $\theta$ and $\theta + 2\pi$ describe the same theory.[1] We will explain that under certain conditions the models above do not posses trivial phases. We show that, as in Yang Mills theory with massless fermions, this is due to 't Hooft anomalies. A notable difference though is that these models are purely bosonic and the associated 't Hooft anomalies are discrete.

We use anomaly inflow arguments to constrain the theories supported on domain walls that appear upon deforming (1). A more detailed analysis of the domain walls, their semi-classical limit and quantum dynamics will be presented elsewhere [14]. We also utilize the global anomalies to constrain the dynamics of these theories at finite temperature and with various chemical potentials.

Various special cases of (1) appear in applications of high energy physics and condensed matter physics. For example, the case of $p = 1, N = 2$ in 3d describes the Néel-VBS transition [4,5] while the case of $p = 1, N = 2$ and $\theta = \pi$ in 2d describes the Haldane model with half-integer spin [15,16]. The case of $p = 1, N = 1$ in 3d is dual to the XY model through the famous particle-vortex duality [17,18]. Both in 2d and 3d these models have a well understood large $N$ limit and nontrivial supersymmetric counter-parts which can be softly deformed. These models are also often studied as toy models for various aspects of four-dimensional Yang-Mills theory (e.g. the 3d $N = 0$ model is nothing but Polyakov's compact Abelian gauge field model [19]).

An interesting general question is under what circumstances the models (1), (2) flow to a conformal field theory. This is equivalent to the question of whether the associated phase transition is first or second order. We will see that sometimes one can use global arguments to prove that the transition must be second order. But even away from these special points where we encounter a conformal field theory, the massive phases of the model are constrained by anomalies and the consequences are often nontrivial.

An example of a case where we can argue that the transition is second order[2] is the 1+1 model with $N = 1$ (i.e. scalar QED in 1+1 dimensions) at $\theta = \pi$. The universality class is

---

[1]The similarity transformation between the theories with $\theta$ and $\theta + 2\pi$ is implemented with the unitary operator $e^{i \int_\gamma A}$, where $\gamma$ parameterizes the space-like slice.

[2]We would like to thank J. Cardy, M. Metlitski, N. Seiberg, and A. Zamolodchikov for comments on this example.

that of the Ising model. The Ising spin field $\Phi$ (which is real valued) would map to the field strength $da$ and the energy operator $\Phi^2$ maps to the mass operator $|\phi^2|$

$$-\frac{1}{4e^2}(da)^2 + \frac{1}{2}da + |D_a\phi|^2 + \lambda|\phi|^4 \longleftrightarrow (\partial\Phi)^2 + \Phi^4 \,.$$

This is remarkably similar to the particle-vortex duality in 2+1 dimensions, which relates a complex field with a complex field coupled to a gauge field. Here we relate a real field with a complex field coupled to a gauge field. This makes sense since the gauge field does not carry degrees of freedom in 1+1 dimensions. It is also worth noting that a similar duality holds for the Schwinger model at $\theta = \pi$ [20], so we in fact have a triality.[3]

For the Abelian Higgs model in 2+1 dimensions with two charged fields of unit charge, there is some evidence that the transition is 2nd order as well. The symmetries of the model away from the putative fixed point are $C \ltimes [SO(2)_{\text{magnetic}} \times SO(3)_{\text{flavor}}] \subset O(2) \times O(3)$. We determine the anomaly of this model. The anomaly is non-trivial even if only a $\mathbb{Z}_2 \subset SO(2)$ is preserved, ie. we allow only even charge monopoles in the Lagrangain, explicitly breaking the $SO(2)$ symmetry to $\mathbb{Z}_2$. The role of $\mathbb{Z}_2 \subset SO(2)$ has been also recently emphasized in [23]. Denoting the $\mathbb{Z}_2$ gauge field by $A$, the anomaly inflow term is expressed in cocycles by

$$\frac{1}{2}\int_4 A \cup w_3(O(3)) \,, \tag{3}$$

where $w_3(O(3))$ is the third Stiefel-Whitney class of the $O(3)$ gauge bundle which combines charge conjugation and flavor symmetry. Therefore, the phases of the model have to either break $O(3)$ (as in the Néel phase) or $SO(2)$ (as in the VBS phase) or be massless (as at the second order transition – but a first order transition is also allowed as far as this analysis goes). In addition, a nontrivial TQFT could saturate the anomaly.[4] A gapped trivial phase where charge conjugation is broken but $SO(2)$ and $SO(3)$ are preserved cannot exist since the anomaly remains nontrivial if we restrict to $SO(2) \times SO(3)$ bundles. Indeed, if we ignore the charge conjugation symmetry (3) reduces to (using the Wu formula [24])

$$\frac{1}{2}\int_4 A \cup w_3(SO(3)) = \frac{1}{2}\int_4 A \cup \frac{1}{2}dw_2(SO(3)) = \frac{1}{2}\int_4 \frac{1}{2}dA \cup w_2(SO(3)) = \int_4 A^2 w_2(SO(3)),$$

using

$$A^2 = \frac{1}{2}dA \mod 2.$$

Note $\mathbb{Z}_2$ cocycles like $w_2(SO(3))$ and $A$ are only closed modulo 2.

This anomaly can also be naturally written also as $\frac{1}{2}\int_4 c(A)w_2(SO(3))$, where $c(A)$ is the Chern class of $A$ considered as a $SO(2)$ connection, more or less just the field strength of $A$. We see that if we ignore the charge conjugation anomaly and assume the $SO(2)$ symmetry is preserved, we find the term written in [25]. It is useful however to keep track of the anomaly involving the charge conjugation symmetry since the domain wall theory in the VBS phase carries a mixed charge-conjugation/$SO(3)$ anomaly which can be written as

$$\frac{1}{2}\int_3 w_3(O(3)) \,. \tag{4}$$

In the absence of charge conjugation symmetry, there would seem to be no obstruction for the domain wall theory to be trivial. Indeed, charge conjugation symmetry pins the coefficient

---

[3] After our paper appeared on the ArXiv, it was pointed out to us that the statement of the triality already existed in the condensed matter literature, see, e.g., [21, 22]. The arguments we provide here give evidence that the transition is second order but we do not rule out all possibilities.

[4] We thank N. Seiberg for proposing a concrete way to do this.

of the term (4) which could otherwise be tuned continuously to zero. Motivated by [11], we discuss the embedding of the $SO(2) \times SO(3)$ anomaly above (3) into $SO(5)$ and $O(4)$ anomalies.

The discussion above of the protected phases of certain Quantum Field Theories leads to powerful non-perturbative constraints on the zero temperature phases of these theories. Typically, however, if we study the theory on a circle, corresponding to turning on finite temperature, the anomalies disappear. Indeed, if we reduce the anomaly (3) on a circle without any chemical potentials, then the anomaly simply vanishes. The phase diagram would therefore contain protected phases at zero temperature but at sufficiently high temperatures the system is disordered with a trivial vacuum. Equivalently, when we study quantum systems on spaces of the form $\mathbb{R}^{d-1,1} \times S^1$, for sufficiently small $S^1$ (and standard thermal boundary conditions), we always expect that all the symmetries are restored. We can think about theories on $\mathbb{R}^{d-1,1} \times S^1$ as being described by local effective field theory on $\mathbb{R}^{d-1,1}$. The effective theory is valid at sufficiently large distances, much larger than the radius of the circle. The statement is that for sufficiently small $S^1$ this effective theory would have a trivial ground state.

This statement has an interesting, well known, loophole. The symmetries of the effective theory on $\mathbb{R}^{d-1,1}$ may descend from standard (0-form) symmetries of the original theory on $\mathbb{R}^{d,1}$ or they could descend from center symmetries of the original theory on $\mathbb{R}^{d,1}$. So the symmetry group of the effective theory $\mathbb{R}^{d-1,1}$ is generally bigger than that of the original theory on $\mathbb{R}^{d,1}$. Symmetries that started their life as standard 0-form symmetries on $\mathbb{R}^{d,1}$ are expected to be restored on $\mathbb{R}^{d-1,1}$ for sufficiently small $S^1$. Perhaps this can be proven to be always the case. But symmetries that descended from center symmetries are not subject to this expectation and in fact they generally prefer to be spontaneously broken for small $S^1$. Here we present some simple examples where one can prove, using anomalies that involve center symmetries, that a trivial ground state cannot exist on $\mathbb{R}^{d-1,1}$ for any value of the radius of the $S^1$. Such systems may have interesting applications. In fact, $SU(N)$ Yang-Mills theory at $\theta = \pi$ has the property that it remains ordered at any temperature [3]. Our examples below are simpler, but they are rather similar in some respects.

Indeed, we show that the Abelian Higgs Models with $p > 1$ (i.e. where the charge of the dynamical particle is bigger than 1) have a 1-form symmetry (i.e. center symmetry) which has a mixed anomaly with the magnetic $SO(2)$ in 2+1 dimensions and a mixed anomaly with time reversal in 1+1 dimensions. Anomalies involving two-form gauge fields (which are the sources for the 1-form symmetry) remain nontrivial upon a reduction on a circle and hence the theory remains ordered even at finite temperature. In 2+1 dimensions, at sufficiently high temperatures, the center symmetry is expected to be broken and hence the response of the free energy to an external charged particle of charge 1 depends on the way the infinite volume limit is taken, analogously to the ambiguity in the response of the free energy to a local perturbation in the external field in the (ordered) ferromagnetic phase.

In 1+1 dimensions the effects of having the fundamental particle have charge $p$ are different. At zero temperature this leads to different superselection sectors which can be thought of as adding stable cosmic strings to the vacuum (the cosmic string is protected by center symmetry). There are therefore multiple superselection sectors in on $\mathbb{R}^{1,1}$ which are not necessarily degenerate in energy. To make the model a little more interesting we imagine that there exists also a heavy charge 1 particle that renders these cosmic strings unstable and removes these superselection sectors. The dynamics is then nontrivial. Quantum effects remove the Higgs phase and the Ising phase transition. Charge conjugation (or alternatively, time reversal) is always spontaneously broken at $\theta = \pi$. Since these models in fact have a mixed time-reversal/center anomaly,[5] the persistence of charge-conjugation breaking at zero temperature is not surprising. This persistent order both at zero and finite temperature is reminiscent of Yang-Mills theory at $\theta = \pi$ [3].

---

[5]This anomaly is only approximate since we introduced heavy charge 1 quarks.

We also discuss briefly the consequences of turning on holonomies on the $S^1$ for background gauge fields. In the context of thermal field theory, this corresponds to chemical potentials. It follows from our analysis of the anomalies that in the 2+1 dimensional Abelian Higgs model (at $p = 1$) with a holonomy for the magnetic $\mathbb{Z}_2$ symmetry, there cannot be a disordered phase regardless of the radius of the $S^1$, i.e. the system remains ordered at any temperature. We therefore see that in the presence of 't Hooft anomalies there are at least two general mechanisms that can guarantee that the theory remains ordered at arbitrary temperature: one-form symmetries or a chemical potential for a standard zero-form symmetry.

The outline of the paper is as follows. We begin with the analysis of 1+1 dimensional Abelian Higgs models in Section 2. We study their anomalies and phases. We also consider the duality with the Ising model and the interesting variation of these models where the fundamental charge is $p > 1$. In section 3 we consider 2+1 dimensional Abelian Higgs models and we analyze their anomalies and phases. We emphasize the role of charge conjugation and determine the anomaly of the domain wall theory. We show that upon various circle reductions one can make contact with the anomalies of the 1+1 dimensional models. We analyze the consequences of the mixed $SO(2)/\text{center}$ anomaly which arises if $p > 1$ and argue that it leads to persistent order everywhere in the thermal phase diagram. Many technical details are collected in several appendices.

## 2 2d Abelian Higgs Models

We begin in 1+1 dimensions with a $U(1)$ gauge field $a$ coupled to $N$ charge $+1$ complex scalars $\phi_i$. The Lagrangian is

$$\mathcal{L} = \frac{1}{4e^2}|da|^2 + \frac{\theta}{2\pi}da + \sum_i |D_a \phi_i|^2 + \lambda \left( \sum_i |\phi_i|^2 \right)^2 . \tag{5}$$

The parameter $\theta$ is $2\pi$-periodic since all the configurations have $\int_\Sigma da \in 2\pi\mathbb{Z}$ for closed $\Sigma$.

An important perturbation that we can add to the Lagrangian is the mass term:

$$\delta\mathcal{L} = M^{ij} \phi_i \phi_j^* , \tag{6}$$

where $M$ may be any Hermitian matrix.

**Symmetries**: We first discuss the symmetries of the model at the massless point $M^{ij} = 0$. There is a manifest $SU(N)$ flavor symmetry rotating the $\phi_i$. However, the element that generates the center of $SU(N)$ acts by a gauge transformation $\phi_i \to e^{\frac{2\pi i}{N}}\phi_i$, hence we should consider it a trivial global symmetry. Taking the quotient of the flavor symmetry by these central elements, we see that all the gauge invariant operators transform under $SU(N)/\mathbb{Z}_N = PSU(N)$. These are the only global symmetries continuously connected to the identity. For us, the discrete symmetries would also be very important. There is a charge conjugation symmetry that acts as

$$C: \quad \phi_i \to \phi_i^* , \qquad a \to -a . \tag{7}$$

This preserves the Lagrangian (5) only at $\theta = 0$ and $\theta = \pi$.

Charge conjugation does not commute with $PSU(N)$ but acts on $PSU(N)$ in a simple way. Indeed, let $U \in SU(N)$ then from the action on the fundamental representation (where the scalars live) we see that $CUC = U^*$.

A generic choice of the mass terms breaks both symmetries. Indeed, $M^{ij}$ can be thought of as a Hermitian matrix in the adjoint of $SU(N)$. Under charge conjugation, $M \to M^*$.[6]

---

[6]Since we can diagonalize $M$, then at least a $U(1)^{N-1}$ symmetry would remain (ignoring some discrete identifications).

However, the diagonal mass $M^{ij} = \delta^{ij}$ respects both $PSU(N)$ and charge conjugation.

**Anomaly at $\theta = \pi$:** Suppose we turn on $SU(N)/\mathbb{Z}_N$ background fields $B$. The total gauge group is an extension

$$U(1) \to U(N) \to PSU(N). \tag{8}$$

This implies a relation of characteristic numbers $N \int da = \int TrF$ where $F$ is the curvature of the total $U(N)$ bundle and the trace is in the fundamental representation. Since this integrated trace may be any $2\pi$ integer on a closed surface, the integral $\int da$ is quantized in "fractional" units of

$$\oint da \in \frac{2\pi}{N}\mathbb{Z}.$$

One can get such "fractional" flux because one can unwind an $N$-wound loop around $U(1) \hookrightarrow U(N)$ through the $SU(N)$ part. There is an invariant of $PSU(N)$ bundles we denote $u_2(B)$, which precisely measures this mod $N$ flux

$$\oint \frac{da}{2\pi} + \frac{u_2(B)}{N} \in \mathbb{Z}, \tag{9}$$

where $u_2 \in H^2(BPSU(N), \mathbb{Z}_N)$ can either be considered as defined by this equation or as the 2-cocycle of the extension (8). For $N = 2$, $PSU(2) = SO(3)$, $SU(2) = Spin(3)$ and so we can identify $u_2$ with the 2nd Stiefel-Whitney class.

Because the $da$ fluxes are quantized in units of $1/N$ in the presence $PSU(N)$ gauge fields, $\theta$ is only periodic under $\theta \to \theta + 2\pi N$. More precisely, from (9) we can read off the transformation:

$$\log Z[\theta + 2\pi, B] - \log Z[\theta, B] = -\frac{2\pi i}{N} \int u_2(B). \tag{10}$$

Note that the right hand side is well defined mod $2\pi i$. It is useful to rephrase the property of the partition function (10) as an anomaly at $\theta = \pi$.

Indeed, charge conjugation at $\theta = \pi$ sends $\theta \mapsto -\pi$ and so necessitates a shift of $\theta$ by $2\pi$ in order to return to the same theory. But we have seen that this shift is nontrivial in the presence of nontrivial gauge fields $B$. Note that $C : B \mapsto -B$ so without anomaly we expect $S_{eff}(B) = S_{eff}(-B)$. Instead, we find

$$S_{eff}(\pi, B) = S_{eff}(-\pi, -B) = S_{eff}(\pi, -B) - \frac{2\pi i}{N} \int u_2(B). \tag{11}$$

In order to cancel this we have to add a fractional contact term to the Lagrangian:

$$\Delta S(B) = \frac{2\pi i}{2N} \int u_2(B). \tag{12}$$

Since it is derived from the curvature, $u_2$ is odd under $C$, so the variation of this term exactly cancels the variation above:

$$S_{eff}(\pi, B) + \Delta S(B) = S_{eff}(\pi, -B) + \Delta S(-B).$$

The problem is that (12) transforms under large $PSU(N)$ gauge transformations. Indeed, $u_2(B)$ is only gauge-invariant modulo $N$. Therefore, adding (12) to the action restores charge conjugation symmetry but spoils $PSU(N)$ gauge invariance. This is very similar to the anomaly of the free fermion in 2+1 dimensions, where we can naively cancel the parity anomaly by adding a half integer Chern-Simons term, but to define such a term requires extra choices (like a 4-manifold filling).

When $N$ is odd this is actually not a problem because we can find an $m$ such that $2m = 1$ mod $N$. Then we may write the counterterm

$$\Delta S = \frac{2\pi i m}{N} \int u_2(B) \,, \tag{13}$$

which is completely gauge-invariant and cancels the $C$-variation of $S_{eff}(\pi, B)$ up to $2\pi i$ integers.

However, when $N$ is even, there is no $PSU(N)$ gauge invariant two-dimensional counterterm that can restore charge conjugation invariance at $\theta = \pi$. We therefore conclude that at $\theta = \pi$ there is a mixed anomaly between charge conjugation and $PSU(N)$ symmetry. Alternatively, we could say that there is a mixed anomaly between time reversal and $PSU(N)$ symmetry. We note that without adding the term (12), the bare action (5) with minimal coupling to $B$ is $PSU(N)$ gauge invariant but not $C$ invariant. The anomaly is only there when both symmetries are considered.

To prove that there is no counterterm, it suffices to show first that the 2d theory coupled to background gauge fields may be defined consistently on the boundary of a 3d theory depending only on the background gauge fields [26]. And second that that 3d theory has a nontrivial partition function on a closed 3-manifold, indicating that this formulation of the theory depends on the choice of 3d bulk with its extensions of the background gauge fields. Indeed, if there was a 2d counterterm, then by Stokes' theorem, all such partition functions would be 1. Again one should draw an analogy with the parity anomaly of free fermions in 2+1D, where the addition of a level 1/2 Chern-Simons term for the background $U(1)$ gauge field requires a choice of 4-manifold filling with extension of the $U(1)$ gauge field. The half-quantized level leads to a dependence on the bulk, measured for example by the partition function on $\mathbb{CP}^2$.

To construct the 3d action, we must also turn on a background gauge field for charge conjugation, which combines with $B$ to form a $PSU(N) \rtimes \mathbb{Z}_2^C$ gauge field we denote $B'$. Combined with the dynamical $U(1)$ gauge field, the total gauge symmetry is now $U(N) \rtimes \mathbb{Z}_2^C$ and there is a class $u_2(B') \in H^2(B(PSU(N) \rtimes \mathbb{Z}_2^C), U(1)^C)$[7] which classifies the central extension

$$U(1) \rightarrow U(N) \rtimes \mathbb{Z}_2^C \rightarrow PSU(N) \rtimes \mathbb{Z}_2^C$$

and twists the magnetic flux quantization for $a$ according to

$$\int \frac{D_{u_1} a}{2\pi} + \frac{u_2(B')}{N} \in \mathbb{Z}$$

analogous to (9), but where now since $a$ is charged under the $\mathbb{Z}_2^C$ part of $B'$, denoted $u_1(B') \in H^1(BPSU(N) \rtimes \mathbb{Z}_2^C, \mathbb{Z}_2)$, we must use the covariant derivative, which may be written as

$$D_{u_1} a = da - a \wedge \Upsilon_1(B),$$

where $\Upsilon_1(B)$ is a flat $U(1)$ connection with holonomy $e^{\int_\gamma \Upsilon_1(B)} = (-1)^{\int_\gamma u_1(B)}$ around closed curves $\gamma$. Note that we wish to consider non-simply connected spacetimes, for which such flat connections are not merely gauge transformations of the trivial connection. In this case, we must work with the covariant derivative. The twisted coefficients $U(1)^C$ in $H^*(B(PSU(N) \rtimes \mathbb{Z}_2^C), U(1)^C)$ indicate cohomology taken with respect to $D_{u_1}$.

Accordingly, to write the $\theta = \pi$ term preserving all global symmetries, we must write it in the combination

$$\int_2 \frac{1}{2} \left( \frac{D_{u_1} a}{2\pi} + \frac{u_2(B')}{N} \right). \tag{14}$$

---

[7]$C$ acts by charge conjugation on the coefficient group, which is identified with the dynamical $U(1)$ gauge group.

We recognize the fractional contact term (12) in the second term.

As with (12), (14) is not invariant under local $PSU(N) \rtimes \mathbb{Z}_2^C$ transformations. One way to measure this is to use Stokes' theorem, noting that if we act by $D_{u_1}$ on the integrand, using $D_{u_1}^2 = 0$ (flatness of $\Upsilon_1$) and $D_{u_1} u_2(B')/N =: u_3(B') \in H^3(BPSU(N) \rtimes \mathbb{Z}_2^C, \mathbb{Z}^C)$, we see that the gauge variations of (14) exactly cancel the boundary variations of the 3d topological term

$$S_{3d}^{anom} = \pi i \int_3 u_3(B') \, . \tag{15}$$

To understand this result, it is useful to specialize to the case $N = 2$. Then we can identify $PSU(2) = SO(3)$, $PSU(2) \rtimes \mathbb{Z}_2^C = O(3)$, and the classes $u_j$ with the corresponding Stiefel-Whitney classes. In particular, $u_3(B') = w_3(B')$ can be understood as the "hedgehog number" of the $O(3)$ gauge bundle. This is defined by considering the adjoint $so(3)$ bundle, which is a 3d real vector bundle on which $C$ acts by $\vec{v} \mapsto -\vec{v}$. A generic section of such a bundle over a 3-manifold has isolated zeros which are imaginatively called hedgehogs. The hedgehog number, or degree, of such a zero is measured by the degree of the direction field associated to a small sphere around the zero, considered as a map $S^2 \to S^2$. A local model for a $+1$ hedgehog is the unit 3-ball with the vector field $\vec{r}$.

One can think of the Néel order parameter of the 1+1d anti-ferromagnetic chain as such a section. This system has an instanton on $S^2$ associated with the degree 1 $SO(3)$ bundle over $S^2$. This bundle has $\int_2 w_2(B') = 1$. The anomaly $w_3/2 = dw_2/4$ indicates there is a $\pi/2$ Berry phase associated with this instanton, which is not compatible with charge conjugation symmetry, which maps this to an anti-instanton with $-\pi/2$ Berry phase. Note that the unit hedgehog $\vec{r}$ on the 3-ball can be "combed" on the surface to reveal not one but *two* instantons on the boundary, related to the well-known fact that the Euler characteristic of the sphere is 2. Only vector fields with even hedgehog number on closed surfaces may be extended to 3-manifold fillings.

Another way to understand the term $u_3(B')$ is to unpack its definition (choosing a gauge for $u_1(B')$)

$$u_3(B') = \frac{1}{N} D_{u_1} u_2(B') = \frac{du_2(B')}{N} - \frac{2u_1(B')u_2(B')}{N} \, . \tag{16}$$

Later we will use this to understand the nontrivial physics on the charge conjugation domain wall.

Anyway, to finish the proof that there is no 2d counterterm which can remedy all this, we need to show that (15) has a nontrivial partition function on some orientable 3-manifold with $PSU(N) \rtimes \mathbb{Z}_2^C$ bundle. For $N = 2$ we can use the 3-manifold $\mathbb{RP}^3 = SO(3)$ and its unique nontrivial $O(1)$ bundle $L$, forming the $O(3)$ adjoint bundle $L \oplus L \oplus L$, which one checks has $\int_{\mathbb{RP}^3} w_3 = 1$. One can think of this bundle as the one whose global sections are spanned by the Pauli matrices. Each of these, upon $2\pi$ rotation about any axis, return to minus themselves. We can mimic this construction for general $N$ by taking a sum of $N^2 - 1$ copies of $L$ to form a $PSU(N) \rtimes \mathbb{Z}_2^C$ adjoint bundle. Note that $w_3$ of this bundle equals $u_3$ mod 2, which is 1 so long as $N^2 - 1$ is odd, ie. when $N$ is even. Indeed, as we said, the existence of the counterterm (13) implies that for odd $N$, the theory (15) has trivial partition functions.

We return to discussing the odd $N$ case. Even though there is no anomaly in the usual sense, the counterterm (13) is discrete, and there is no way to put a continuous parameter in front of it. If we add it to the action (5) as written, we get a theory which is anomaly-free at $\theta = \pi$ but has an anomaly at $\theta = 0$! There is no local counterterm which can be written for all $\theta$ which preserves the symmetry at both $\theta = 0$ and $\theta = \pi$. In general, when we discuss anomaly in the abstract we usually consider variations which cannot be canceled by local counterterms and instead are canceled by a non-trivial bulk counterterm. Sometimes this bulk

counterterm is globally trivial, but locally nontrivial, like $w_3$ mod 2 or the second Chern number $F \wedge F$ (by "globally trivial" here we mean that one can put a continuous parameter in front of these terms and so one can continuously remove these terms by 2D counterterms). Such terms always contribute a discrete-coefficient non-trivial boundary term, eg. the Chern-Simons term or our counterterm (13). These may be considered "secondary anomalies" analogous to the mathematical notion of secondary characteristic classes like the Chern-Simons invariant. When there is a continuous family of theories with different secondary anomalies at different points, counterterms defined on the entire family can only move around the secondary anomalies. There is no way to have all the classical symmetries on the whole family.

Anomaly inflow can get around this obstruction. Indeed, $u_3(B)$ is an integer class, so it make sense to write the bulk counterterm with a continuous parameter

$$i\theta \int_3 u_3(B). \tag{17}$$

At $\theta = 0$ this term is trivial and at $\theta = \pi$ this term is our counterterm (15). Thus, adding this bulk term to (5) for any $N$ fixes the symmetries at both $\theta = 0$ and $\theta = \pi$. This works whenever the anomaly polynomial has an integer lift and so admits a continuous coefficient but the coefficient is pinned by a discrete symmetry. We therefore conclude that even though there is no anomaly in the usual sense at $\theta = \pi$ for odd $N$, the anomaly still exists in a slightly weaker sense and the physical consequences are the same as if there were a standard anomaly. More on this point was recently discussed by one of us in [27].

Finally, we can imagine turning the fields $B$ into dynamical fields. Then we are studying the 1+1 dimensional system of a $U(N)$ gauge field coupled to a scalar in the fundamental representation. The periodicity of $\theta$ is now $2\pi N$ and the theory at $\theta = \pi$ is explicitly not CP invariant. If $N$ is odd we can add a local counterterm (13) which preserves charge conjugation symmetry at the $\theta = \pi$ point but breaks it at $\theta = 0$. For general $N$, we may couple to a 3D bulk carrying the topological term (17) and this preserves charge conjugation symmetry at $\theta = \pi$ and $\theta = 0$.

Let us now turn to discussing the special cases $N = 0$ and $N = 1$ where the $PSU(N)$ symmetry is trivial and hence a separate discussion is needed for completeness. We will then return to the models with $N > 1$ and discuss their various phases (and briefly also their domain walls).

## 2.1 $N = 0$

With no matter, the model is free. It is described by the Lagrangian

$$\mathcal{L} = \frac{1}{4e^2} da \wedge \star da + i\frac{\theta}{2\pi} da . \tag{18}$$

There are no propagating degrees of freedom in $\mathbb{R}^{1,1}$ but upon placing the theory on a circle of radius $R$ there are propagating degrees of freedom associated with the holonomy $q = \int_{S^1} a$, which is $2\pi$ periodic due to large $U(1)$ gauge transformations. The effective action of $a$ in the compactified model is

$$\mathcal{L} = \frac{1}{2e^2 R} \dot{q}^2 + i\frac{\theta}{2\pi} \dot{q} . \tag{19}$$

This model is the one studied in appendix D of [3] and it has a two-fold degenerate ground state at $\theta = \pi$. Otherwise it has a unique ground state and the gap scales like $e^2 R$, which means that the energy density of the excited state is larger by $e^2$ than in the vacuum.

The 2D model (18) has a continuous 1-form center symmetry, which is typically accidental (though a subgroup of it may be preserved in the microscopic theory as we will see). The

corresponding classical current is the point operator $\star da$, using the 2D Hodge $\star$. Its correlation functions are independent of where it is inserted due to the Maxwell equation of motion $d \star da = 0$ in the absence of dynamical charges. The center symmetry acts on the gauge field $a$ by shifting it by a flat connection. This 1-form symmetry can be coupled to a background two-form gauge field $K$ by adding $K \wedge \star da + K \wedge \star K$ to the Lagrangian (see Appendix A). This term cancels the gauge variation of the kinetic term, but under a gauge transformation $a \mapsto a + \lambda$, the $\theta$ term has a variation

$$i\frac{\theta}{2\pi} \int_2 d\lambda \,, \tag{20}$$

which may be non-zero, since $\lambda$ is an arbitrary $U(1)$ connection. There is no 2D counterterm that can be assembled out of $K$ that will also be $C$ invariant at $\theta = \pi$. However, there is a special 3-cocycle, the Dixmier-Douady-Chern class $c_3(K) \in H^3(B^2 U(1), \mathbb{Z})$ which may be used to construct the bulk counterterm

$$-i\theta \int_3 c_3(K). \tag{21}$$

Up to torsion contributions, which are important on nonorientable 3-manifolds, this is equivalently $-i\theta/2\pi \int_3 dK$. Under a large gauge transformation, this term varies in precisely the right way to cancel the above variation (see Appendix A).

We summarize that a 1+1 dimensional counter-term that cancels (20) and respects charge conjugation symmetry does not exist. As a result, the free model (18) at $\theta = \pi$ has a mixed anomaly between this one form symmetry and charge conjugation. Indeed, with a charge conjugation background turned on, since the current $\star da$ is $C$-odd, $K$ is as well, so $K$ is promoted to the $BU(1) \rtimes C$ gauge field $\hat{K}$.[8] Then we must promote $c_3(K)$ to $c_3(\hat{K})$, which is only defined modulo 2, and defines a non-trivial anomaly class in $H^3(B[BU(1) \rtimes C], U(1))$.

When we compactify on a circle, we get an ordinary 1-form $U(1)$ gauge field in the remaining direction

$$A = \int_{S^1} K.$$

This gauge field couples to the 0-form current $\star dq$ (this is the 1D Hodge $\star$) corresponding to the $U(1)$ shift symmetry $q \mapsto q + \alpha$ of (19). If the 3D bulk is also the form of a product $M_2 \times S^1$, then we can rewrite the bulk anomaly inflow term (21) as

$$i\theta \int_2 c_2(A).$$

Charge conjugation acts by $C : A \mapsto -A$, so we can think of the combined symmetry as $U(1) \rtimes C = O(2)$ and consider the combined $U(1)$ and charge conjugation $O(2)$ gauge field $\hat{A}$. Then the $\theta = \pi$ term can be written

$$\pi i \int_2 w_2(\hat{A}) \,. \tag{22}$$

This anomaly at $\theta = \pi$ explains the two-fold vacuum degeneracy and how the $O(2)$ symmetry becomes extended to $Pin^+(2)$, the extension classified by $w_2 \in H^2(BO(2), \mathbb{Z}_2)$, as discussed in [3].

The two degenerate ground states $|0\rangle$ and $|1\rangle$ of (19) in the winding number basis are exchanged by charge conjugation, so charge conjugation must also be spontaneously broken in (18) on $\mathbb{R}^{1,1}$, even though the underlying model is free. Of course, this is due to the fact that

---

[8]This must be understood as a gauge field for a 2-group. See [28] for details on these objects and other physical applications.

the electric field could be $\pm\frac{1}{2}$ for $\theta = \pi$. In the absence of charged particles, the degeneracy cannot be lifted even when the theory is compactified. From the point of view of the quantum mechanics, the degeneracy is there because the theory has a global 't Hooft $O(2)$ anomaly. The anomaly inflow term simply consists of the 2nd Stiefel-Whitney class of $O(2)$.

We end this section with a discussion of the 1-form symmetry of the gauge theory and how it relates to the (zero-form) shift symmetry of $q$. Suppose a 1+1D theory with the space taken to be a circle $S^1$ has a symmetry operator $Q$. We can always think of such a theory as a possibly complicated QM model. Then the Hilbert space can be decomposed into different representations $r$ of $Q$

$$\mathcal{H} = \bigoplus \mathcal{H}_r \ . \tag{23}$$

There are typically operators which carry charge under $Q$ and connect the different sectors. However, it may sometimes happen that no such operators come from local operators on $\mathbb{R}^{1,1}$, that they really need to wrap the circle somehow. Then in the flat space limit, the $\mathcal{H}_r$ which cannot be connected by local operators in $\mathbb{R}^{1,1}$ become by definition different superselection sectors. One can often diagnose the superselection sector by measuring the expectation value of a local operator.

One situation in which it is *guaranteed* that the $\mathcal{H}_r$ are different superselection sectors for all $r$ is when $Q$ is a 1-form symmetry. This is because no local operators in $\mathbb{R}^{1,1}$ carry charge under it, by definition. In the theory (18) there is a $U(1)$ 1-form symmetry and the vacua with different $E_k = \frac{1}{2\pi}\theta + k$ carry charge $k$ under $U(1)$. Therefore, no local operators can mix them. We can measure the point operator $\star da$ in order to detect $k$. We can similarly measure the energy density, which is $E_k^2$. A useful intuitive picture is that 1-form symmetries act on strings, namely, objects of dimension 1 in space. But in 1+1 dimensions a string is Poincaré invariant since it is a space-filling object. Therefore, adding a string to the vacuum leads to a new superselection sector if the string is stable (i.e. if the 1-form symmetry is unbroken). We can therefore think of the parameter $k$ as counting cosmic strings, electric field lines going from $-\infty$ to $\infty$.

If we had a massive particle with unit $a$ charge, then this would break the 1-form symmetry explicitly and there would be local operators, eg. the number operator of the particle, which witness tunneling between the different superselection sectors. Intuitively, the cosmic string can end on a unit $a$ charge, so they can decay by creating particle-antiparticle pairs. This way, the unit $a$ charge is a domain wall between different superselection sectors.

Suppose instead the theory only has heavy dynamical particles with even $a$ charge. Then cosmic strings can only decay in pairs, and so a $\mathbb{Z}_2$ one form symmetry remains with two distinct superselection sectors corresponding to $k$ even and $k$ odd. The two sectors are exactly degenerate in energy at $\theta = \pi$ (even when the theory is compactified on a circle). This is a reflection of the fact that the anomaly class of the 0+1D theory $\frac{1}{2}w_2(\hat{A})$ remains non-trivial all the way down to the subgroup $\mathbb{Z}_2 \times \mathbb{Z}_2 \subset O(2)$ generated by

$$R = \begin{pmatrix} -1 & 0 \\ 0 & -1 \end{pmatrix},$$

$$C = \begin{pmatrix} 1 & 0 \\ 0 & -1 \end{pmatrix}.$$

The "rotation" symmetry $R$ is present only so long as the particles all have even $a$ charge. It descends from the $\mathbb{Z}_2$ subgroup of the original 1-form symmetry. We will use this observation in subsection 2.3.

## 2.2  $N = 1$

This can be viewed as 1+1 dimensional scalar QED, or the 1+1 dimensional Abelian Higgs model. Some of its basic properties are described in [29]. The Lagrangian is

$$\mathcal{L} = \frac{1}{4e^2}|da|^2 + i\frac{\theta}{2\pi}da + |D\phi|^2 + m^2|\phi^2| + |\phi|^4 \, . \tag{24}$$

Here we make a few qualitative observations about it. The model admits a charge conjugation symmetry at $\theta = 0$ and $\theta = \pi$. This model does not have a center 1-form symmetry since it has a particle of charge 1. The model at $\theta = 0$ presumably always has an unbroken charge conjugation symmetry, and a trivial and gapped vacuum.

Let us consider the $C$-invariant point $\theta = \pi$ with large positive $m^2 \gg e^2$. In this case, the model is approximately described by the free gauge field model (18). The free model breaks charge conjugation spontaneously with two degenerate ground states related by $C$. These are two different superselection sectors in flat space (and as we explained, the degeneracy remains on a circle due to the center symmetry of the free $U(1)$ model). Now we need to consider the corrections due to the massive particle. Integrating out the heavy scalar leads to corrections of two types to the quantum mechanical model (19).

1. Already in $\mathbb{R}^{1,1}$, integrating out the heavy particle leads to irrelevant operators of the form $|da|^n/m^{2n-2}$. These lead to corrections to the kinetic term of the quantum-mechanical degree of freedom $\dot{q}$. However, since this does not break the $O(2)$ symmetry, the anomaly (22) remains and hence the two-fold ground state degeneracy is unaffected by these corrections even after a circle compactification.

2. Upon compactification the scalar particles can propagate across the circle, with probability scaling like $e^{-2\pi mR}$. These processes break the continuous shift symmetry of $q$ completely[9] but of course $q$ remains a $2\pi$ periodic variable. Hence, we expect a potential of the form (see Appendix B)

$$V(q) \propto R^{-1}e^{-2\pi mR}\cos(q) + \cdots \, . \tag{25}$$

   This breaks the $O(2)$ symmetry to the $C$ subgroup $q \mapsto -q$ and nothing remains of the anomaly (22). Hence, the two-fold degeneracy of the ground state is lifted by an exponentially small term $e^{-2\pi mR}$. In the de-compactification limit this term disappears and we have two degenerate vacua.

To summarize, in $\mathbb{R}^{1,1}$ we have two ground states for $m^2 \gg e^2$ at $\theta = \pi$. Charge conjugation is spontaneously broken. As typical in interacting field theories, upon a circle compactification the degeneracy is removed by an exponentially small term, as we have seen above.

We can fix $m^2 \gg e^2$ and vary $\theta$. The ground state is non-degenerate at $\theta \neq \pi$ and becomes doubly degenerate at $\theta = \pi$ in the infinite space limit $\mathbb{R}^{1,1}$. This is characteristic of a first order phase transition, where an excited state at $0 < \theta < \pi$ is becoming less energetic as we tune towards the phase transition, eventually crossing the ground state at $\theta = \pi$ to become the new ground state for $\pi < \theta < 2\pi$.

Now we consider the opposite limit, with large negative mass squared $-m^2 \gg e^2$. The potential forces a nonzero vev for $\phi$ so the gauge field $a$ is Higgsed. We can arrange the limit so that the gauge field is Higgsed at energies much higher than those where the system becomes strongly coupled. Charge conjugation is manifestly preserved in the Higgs phase and the $\theta$ dependence is subleading because the photon is so massive. Therefore there is a single vacuum for all $\theta$.

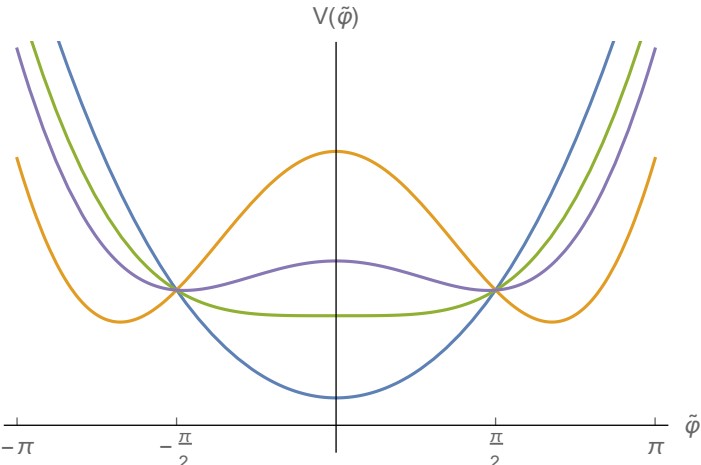

Figure 1: The potential of $\tilde{\varphi}$ as the strength of instanton effect is varied at $\theta = \pi$ in the $p = 1, N = 1$ model. As one moves away from the deep Higgs regime the number of minima changes from one to two, resembling an Ising transition.

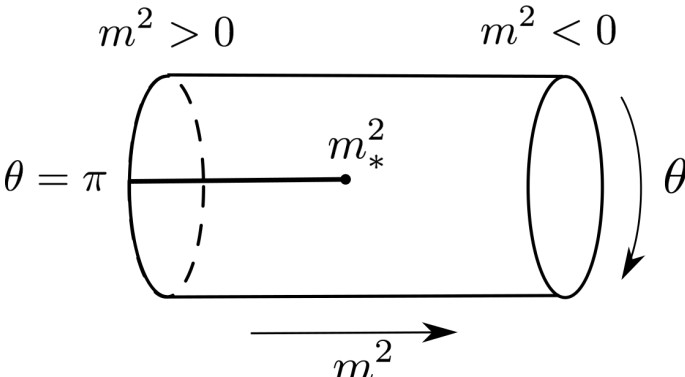

Figure 2: The phase diagram of 1+1 scalar QED with one charge 1 scalar. The semi-infinite line at $\theta = \pi$ represents a first order transition which ends at some critical $m_*^2$ with a second order Ising transition.

We can look at the Higgs phase of the theory in greater detail. In the Higgs phsae there exist vortex instanton configurations. As we move away from the deep Higgs regime we should also consider the corrections due to vortex instantons. Recall in the extreme deep Higgs regime, the radial mode of the scalar is super massive compared to the angular mode and the gauge field and therefore decouples in this limit. The system can be approximately described by the Stückelberg action

$$\mathcal{L} = \frac{1}{4e^2}|da|^2 + i\frac{\theta}{2\pi}da + \frac{1}{2}|d\varphi + a|^2, \tag{26}$$

where $\varphi \sim \varphi + 2\pi$ is the phase of the scalar field. It is convenient to use the $2\pi$-periodic dual variable $\tilde{\varphi}$, related to $\varphi$ by

$$\star d\tilde{\varphi} = d\varphi. \tag{27}$$

---

[9]The shift symmetry originated from the 1-form symmetry in 1+1 dimensions. The existence of charge 1 particles breaks this symmetry completely.

In terms of the dual variable, the action becomes (see appendix C)

$$\mathcal{L} = \frac{1}{4e^2}(da)^2 + \frac{1}{8\pi^2}(d\tilde{\varphi})^2 + \frac{i}{2\pi}\tilde{\varphi}da\,. \tag{28}$$

Note that $\theta$ has disappeared. Indeed, it was removed by a change of variables where we simply shifted $\tilde{\varphi}$. This is of course the standard fact that in the deep Higgs phase the theta dependence is weak and arises due to instantons, as we will see below. The vacuum of the system (28) is located at $\tilde{\varphi} = 0$ and integrating out $a$ gives a quadratic mass term to $\tilde{\varphi}$. When vortex instantons are taken into consideration a new term is introduced to the effective potential (see Appendix E for a derivation)

$$V_{\text{inst}}(\tilde{\varphi}) = -2m_a^2 e^{-S_0}\cos(\tilde{\varphi} + \theta)\,, \tag{29}$$

where $S_0$ is the action of a vortex instanton. At $\theta = 0$, the number of ground states remains one as the strength of instanton effect is varied because $\tilde{\varphi} = 0$ is also a minimum of $V_{\text{inst}}$. However at $\theta = \pi$, $\tilde{\varphi} = 0$ is a maximum of $V_{\text{inst}}$. As we move away from the deep Higgs phase, $V_{\text{inst}}$ becomes more prominent. There will be a point where the cosine potential overcomes the quadratic mass term and the system develops two ground states, similar to a second order Ising phase transition (see Fig. 1).

We can imagine the space of theories, labeled by $m^2, \theta$ as an infinite cylinder (Fig. 2). There is a first order transition line that starts at $\theta = \pi$ and large positive $m^2$, but for large neative $m^2$ there is no transition. Thus, something interesting must happen in between to the transition. The simplest situation is that the first order line simply ends at some special $m_* \sim e^2$. This would be a second order 1+1 Ising transition. However, it is not the only possibility. For instance, it could separate into two first order lines which travel around the cylinder and meet each other at $\theta = 0$, in a way preserving $C$ and $T$ symmetries. Unless we can rule out any interesting behavior $\theta \neq \pi$, we cannot be certain that the situation is not more complicated that a 2nd order Ising transition. This situation should be compared to the Schwinger model, see e.g. [20]. There were some attempts to study the theory (24) on the lattice, e.g. [30], but the conclusions were not definite enough to confirm or exclude our prediction of an Ising transition. It would be very nice to figure out exactly what happens to the transition line at intermediate $m^2$.

Let us proceed as though the simple situation occurs and we have an Ising critical point where the first order line ends. In this case, the mapping of operators between the Ising model and the quantum critical point above can be worked out from the mapping between the Higgs phase of the gauge theory and the disordered phase of the Ising model. The Higgs phase order parameter is $|\phi|^2$ so it gets identified with the energy operator $\epsilon$, the order parameter of the disordered Ising phase. This leaves the spin field $\sigma$ to be identified with $\star da$, so in summary:

$$\star da \leftrightarrow \sigma\,, \qquad |\phi|^2 \leftrightarrow -\epsilon\,.$$

Both the Abelian Higgs model and the Ising model are free of 't Hooft anomalies. Yet, approximate symmetries and their anomalies play a crucial role in establishing the different phases of the model. For example, for $m^2 \gg e^2$ charge conjugation symmetry is broken essentially because the model has an approximate center symmetry and a mixed 't Hooft anomaly with charge conjugation, as in the free QED model.

## 2.3 The Abelian Higgs Model with $p > 1$

Here we study the model (24) where the fundamental charge is $p$. Therefore, the covariant derivative is now defined as $D\phi = d\phi + ipa\phi$.

In the case that the fundamental charge is $p$, there is a $\mathbb{Z}_p$ center symmetry in the problem. The operators that are charged under this symmetry are Wilson loops

$$W_k = e^{ik\int a}\,, \qquad k = 0,..,p-1\,. \tag{30}$$

(We can also describe the $\mathbb{Z}_p$ charge operators. They are simply local operators $O_m$, $m = 0,..,p-1$ which have the following equal-time commutation relation with the Wilson loops $O_m W_k = e^{2\pi ikm/p} W_k O_m$. Hence, the operators $O_m$ create flux $1/p$.)

The Wilson loops (30) are not screened since there are no appropriate dynamical charges in the theory. Acting with these Wilson loops on the vacuum we add a cosmic string which is stable mod $p$. So these are going to be the $p$ different superselection sectors which are not necessarily degenerate in energy. The Hilbert space of the theory depends on $\theta$ and we can write it as a direct sum decomposition of superselection sectors according to the charges under $\mathbb{Z}_p$

$$\mathcal{H}_\theta^p = \bigoplus_{k=0}^{p-1} \mathcal{H}_{\frac{\theta}{p} + \frac{2\pi k}{p}}\,. \tag{31}$$

This decomposition was already written down in [31], see also [32,33]. Here $\theta$ and $k$ come in such a combination because the Wilson loops (30) change $\theta$ by $2\pi k$. In other words, $\theta \to \theta + 2\pi$ just shuffles the $p$ superselection sectors. So the Hilbert space of the charge $p$ theory is made of $p$ copies of $\mathcal{H}_{\theta'}$ at different values of $\theta' \sim \theta' + 2\pi$ and $\mathcal{H}_{\theta'}$ is isomorphic to the Hilbert space of the $p = 1$ theory with theta angle $\theta'$. Our first task is to compute the energy density in these different superselection sectors in the various phases of the theory.

This is rather straightforward in the phase of the theory with $m^2 \gg e^2$. There the model is well approximated by a free $U(1)$ gauge field, where the energy densities in these superselection sectors are proportional to

$$\left(\frac{1}{2\pi}\theta + k\right)^2\,.$$

Tunneling is possible between states where $k$ and $k'$ are the same mod $p$ with the pair production of charge $p$ particles. So all the configurations with electric field less than or equal to $p/2$ are stable. Only if $\theta = \pi$ then the lowest lying superselection sector is degenerate.

Now let us turn to analyzing the Higgs phase of the theory. We start from the deep Higgs phase, where $-m^2 \gg e^2$. The scalar $\phi$ condenses but since it has charge $p$, a $\mathbb{Z}_p$ gauge symmetry remains unbroken and we find a $\mathbb{Z}_p$ gauge theory at low energies. We begin with the approximation where the radial mode is completely decoupled. Then we have the Stückelberg action

$$\mathcal{L} = \frac{1}{4e^2}|da|^2 + i\frac{\theta}{2\pi}da + \frac{1}{2}|d\varphi + pa|^2\,. \tag{32}$$

Here the field $\varphi$ is $2\pi$ periodic and transforms under gauge transformations as usual

$$a \to a + d\lambda\,, \qquad \varphi \to \varphi + p\lambda\,.$$

The theory is quadratic and we can solve it exactly. After a change of variables,

$$\star d\tilde{\varphi} = d\varphi$$

the Stückelberg action becomes (See Appendix C)

$$\mathcal{L} = \frac{1}{4e^2}(da)^2 + \frac{1}{8\pi^2}(d\tilde{\varphi})^2 + \frac{ip}{2\pi}\tilde{\varphi}da\,. \tag{33}$$

In this description it is clear that there is a global $\mathbb{Z}_p$ 0-form symmetry generated by $\tilde{\varphi} \to \tilde{\varphi} + 2\pi/p$. This is the standard $\mathbb{Z}_p$ 0-form symmetry of the $\mathbb{Z}_p$ gauge theory in 1+1

dimensions. In addition, the $\mathbb{Z}_p$ 1-form symmetry is manifest; if we construct the local operator for which $\tilde{\varphi}$ winds by $2\pi$ around some point, we find that this local operator carries charge $p$, so we can identify it with $\varphi$.

Finally, note that (32) has a $\theta$ parameter while after the duality transformation to (33) the $\theta$ parameter has disappeared. Indeed, we can shift $\tilde{\varphi}$ to eliminate $\theta$. This is a reflection of the usual statement that $\theta$ does not matter much in the deep Higgs phase. In particular, in this approximation, there is always a charge conjugation symmetry $da \to -da$ accompanied by $\tilde{\varphi} \to -\tilde{\varphi}$. The model (33) has $p$ degenerate ground states, which can be viewed as arising from spontaneous breaking of the $\mathbb{Z}_p$ 0-form symmetry $\tilde{\varphi} \to \tilde{\varphi} + 2\pi/p$.

To gain some more intuition for this problem imagine putting the theory on a large circle of radius $R$. We choose a gauge where $A_1$ is constant and $\int A_1 = q(t)$, and thus $q$ is a $2\pi$ periodic quantum mechanical variable. Then, after integrating over the circle the Lagrangian reduces to (see Appendix D):

$$\mathcal{L} = \frac{\dot{q}^2}{2e^2 R} + \frac{R}{8\pi^2}(\dot{\tilde{\varphi}}_0)^2 + \frac{ip}{2\pi}\tilde{\varphi}_0 \dot{q} + \frac{R}{8\pi^2}\sum_{k \neq 0}\left[(\dot{\tilde{\varphi}}_k)^2 + (4\pi^2 R^{-2}k^2 + p^2 e^2)\tilde{\varphi}_k^2\right] .$$

The spectrum of the $\tilde{\varphi}_{k \neq 0}$ modes is the standard direct sum of Harmonic oscillators of a massive 1+1 dimensional boson of mass $m^2$. The system $\tilde{\varphi}_0, q$ defines a Landau problem on a torus with $p$ units of magnetic flux. The ground state is therefore $p$-fold degenerate. The higher Landau levels are separated by a gap that scales like $pe$. We therefore have $p$ distinct ground states on the circle. In the approximation that the radial mode is decoupled, these $p$ ground states are exactly degenerate. They can also be described in terms of the orignal variables (32). There, imagine putting the theory on a large spatial circle. We can minimize the action by choosing $A = \frac{k}{p}$, $\partial\varphi = k$ for some integer $k$. Large gauge transformations take $k \to k + p$ and hence we have $p$ distinct ground states (Fig. 3) in which there is a condensation of winding modes. On $\mathbb{R}^{1,1}$, these ground states preserve charge conjugation symmetry. Wilson loops are the domain walls between these vacua. The Wilson loops are the order parameters for the 1-form symmetry (which is present also when the radial model is included).

The $\mathbb{Z}_p$ 0-form symmetry of the model (33) is however removed by quantum corrections (instanton vortices) and so is the vacuum degeneracy. Therefore the $\mathbb{Z}_p$ 0-form symmetry is an artefact of the Stückelberg action, which is obtained if one ignores the dynamics of the radial mode.

Let us imagine a configuration on the cylinder with $\int F = 2\pi/p$. The holonomy would change by $\Delta\int A = 2\pi/p$. Hence, if such configurations have a finite action on the cylinder, then they can be viewed as instantons which lift the $p$-fold degeneracy. The proliferation of these vortex-instantons would break the $\mathbb{Z}_p$ symmetry of the Abelian Higgs model and generate a potential which is local in terms of the variable $\tilde{\varphi}$ (see Appendix E)

$$V(\tilde{\varphi}) = -2m_a^2 \, e^{-S_0}\cos(\tilde{\varphi} + \theta/p) + \cdots . \tag{34}$$

The degeneracy is therefore slightly lifted. In fact, due to the potential (34) there is exactly one ground state for $\theta \neq \pi$ (Fig. 4) and two ground states for $\theta = \pi$ (Fig. 5). The two ground states for $\theta = \pi$ are related by charge conjugation.

An analogous situation takes place in the Schwinger model with the fundamental fermion having charge $p$ under the gauge symmetry. In the massless case the model is equivalent to (33). The $\mathbb{Z}_p$ 0-form symmetry in this cases corresponds to discrete axial rotations which shift $\theta$ by $2\pi$. A small mass term for the fermions would induce the term (34). There is therefore one ground state for $\theta \neq \pi$ and two ground states if $\theta = \pi$. This is reminiscent of the physics of softly broken SYM theory, with the main difference being that here the higher energy vacua are exactly stable because they are protected by the center symmetry.

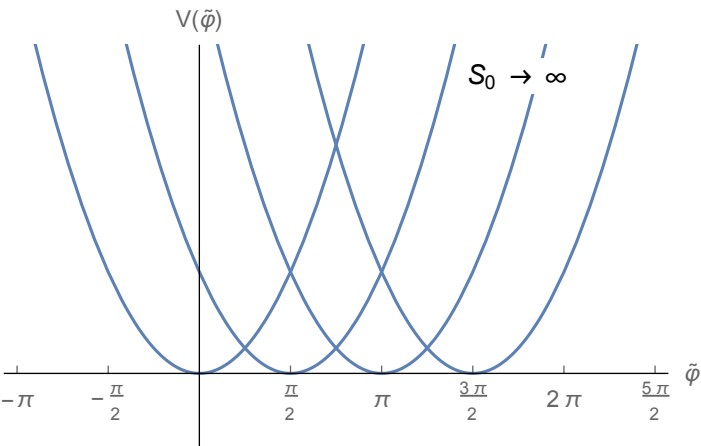

Figure 3: Potential energies of the $p$ superselection sectors in the Higgs phase. Here the action $S_0$ of a vortex instanton is taken to be infinity and the instanton effect is turned off. We have $p$ vacua ($p = 4$) degenerate exactly in energy.

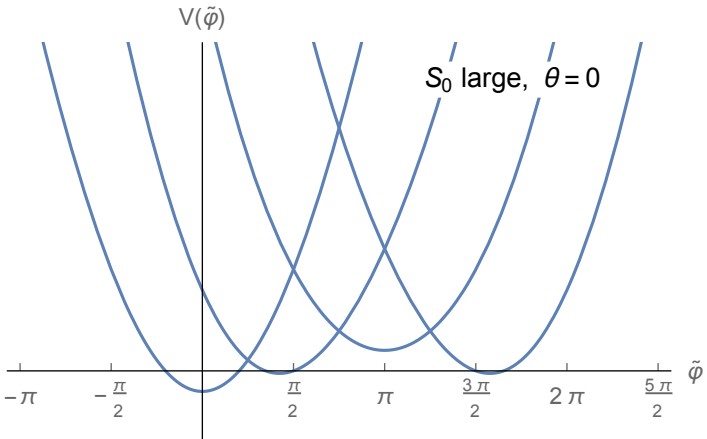

Figure 4: Potential energies of the $p$ superselection sectors in the Higgs phase at $\theta = 0$. Here $S_0$ is taken large but finite. The instanton vortices lift some of the degeneracies of the $p$ sectors, leaving only one sector with the lowest energy density.

To summarize, we see that the model with $p > 1$ has $p$ superselection sectors, of which the lowest energy one is degenerate only if $\theta = \pi$. In order to remove the higher energy supeselection sectors we can now imagine that we add a very heavy charge 1 particle. This allows the strings with the higher energy density to decay via pair creation of charge 1 particles and it removes the other spurious superselection sectors.

We can thus view our analysis here as pertaining to the Abelian Higgs model with one particle of charge $p$ and one heavy particle of charge 1. We see that as we vary the mass of the charge $p$ particle, at $\theta \neq \pi$ there is always a single ground state and at $\theta = \pi$ there is an exact two-fold degeneracy associated with the spontaneous breaking of charge conjugation symmetry. This is rather surprising: we see that the Ising transition in the $p = 1$ model is removed! For $p > 1$ there is no longer a phase with unbroken charge conjugation invariance.

Let us understand this surprising conclusion from the point of view of the anomalies in the system.

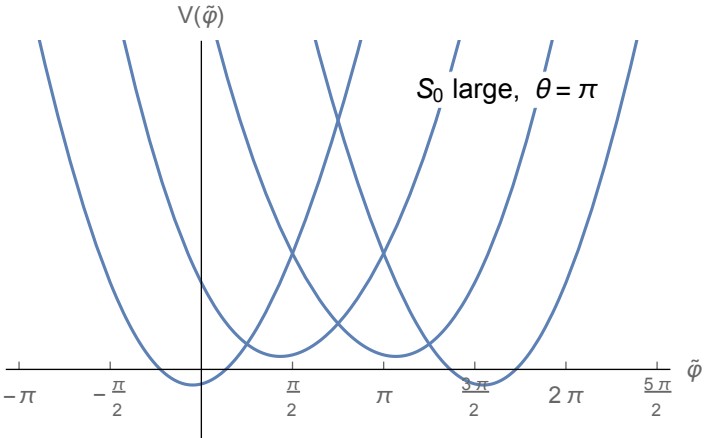

Figure 5: Potential energies of the $p$ superselection sectors in the Higgs phase at $\theta = \pi$. Here $S_0$ is taken large but finite. The instanton vortices lift some of the degeneracies of the $p$ sectors but there is a two-fold degeneracy in sectors with lowest energy densities.

First, we use the center symmetry to couple the system to a background two-form $\mathbb{Z}_p$ gauge field, $K$. As in subsection 2.1, The integral $\int da$ is then quantized in units of $2\pi/p$ such that

$$\int_2 da + 2\pi K/p \in 2\pi\mathbb{Z},$$

for all closed surfaces. Thus, under a transformation $C : \theta \mapsto -\theta$ followed by $\theta \mapsto \theta + 2\pi$, the action is not invariant but

$$\delta S = \frac{2\pi i}{p} \int K .$$

Again, as in subsection 2.1, we can re-interpret this equation as a mixed anomaly between charge conjugation at $\theta = \pi$ and the 1-form $\mathbb{Z}_p$ symmetry. Indeed, we could cancel this with a 2D counterterm

$$\frac{2\pi i}{2p} \int K,$$

but $K$ is only defined mod $p$. If $p$ is odd, then we can find an integer $m$ such that $2m = 1$ mod $p$. Then we may write the counterterm

$$\frac{2m\pi i}{p} \int K , \tag{35}$$

and there is no anomaly. However, if $p$ is even, then there is no obvious way to write a 2D counterterm. We can instead just take $d$ and put the system on the boundary of a 3D bulk theory with topological action

$$\pi i \int_2 \frac{dK}{p} . \tag{36}$$

Since $\frac{dK}{p} \neq 0 \in H^3(B^2\mathbb{Z}_p, U(1)^C)$, it is impossible to write this term locally on the 2D boundary, so we have a genuine anomaly. Note that this contributes just a sign to the path integral.

This anomaly is indeed merely a restriction of (21) to the case that the 1-form symmetry is $\mathbb{Z}_p \subset U(1)$. Note that while the anomaly has a 2D counterterm (35) for $p$ odd (just as our

discussion for $N$ odd above) there is no way to put a continuously varying coefficient in front of it which could interpolate between a $C$-invariant theory at $\theta = 0$ and a $C$-invariant theory at $\theta = \pi$. Thus, when considering theories where we vary $\theta$, there is still in a sense an anomaly. From our discussion above, we see that the anomaly is saturated by persistent breaking of charge conjugation. Indeed, a discrete 1-form symmetry cannot be broken in 1+1 dimensions so it is therefore not entirely surprising that charge conjugation must be always broken.

## 2.4  $N > 1$ **Models**

For $\theta = 0$ these models are believed to be always gapped, with a trivial, non-degenerate vacuum. However, at $\theta = \pi$ there is anomaly inflow from the 3d term (15). This means that the ground state either has to break the global symmetries (i.e. charge conjugation[10]) or there would be a nontrivial theory in the infrared (which may be either gapless or topological). For $N = 2$ the conformal field theory that we find as we vary the parameters is a $SU(2)_1$ WZW model [34]. For $N > 2$ it is believed that charge conjugation is always spontaneously broken [21, 35]. This is certainly the case for sufficiently large $N$ [36]. Therefore, the vacuum is two-fold degenerate and the discrete $u_3$ anomaly is saturated by the spontaneously broken charge conjugation symmetry.

We can discuss the domain wall in this theory. The anomaly is obtained by integrating (15) over the dimension orthogonal to the wall in the presence of a nonzero charge conjugation gauge field, done using (16). It is a nontrivial quantum mechanical theory at the boundary of the two-dimensional topological term

$$\frac{2\pi i}{N} \int_2 u_2(B) \,.$$

Therefore, this is a quantum mechanical model with a projectively realized $PSU(N)$ symmetry which is centrally extended (due to a quantum anomaly) to $SU(N)$. The smallest such representation we can have is the fundamental representation of $SU(N)$ and so the ground state of this quantum mechanical model is at least $N$-fold degenerate.

To gain a little more intuition for the quantum mechanical model on the domain wall, consider $N = 2$. The low energy theory on the domain wall in this case can be understood rather intuitively. Indeed, a simple example of a particle with this anomaly is a unit electric charge on $S^2$ with a unit magnetic flux. The ground state is two-fold degenerate due to the fact that the spin of the monopole-electron system is half integer, and the ground state is in the fundamental representation of $SU(2)$. This model was discussed from this point of view, e.g. in [25]. (It can also be viewed as an orbifold with a Chern-Simons term of particle on the $SU(2)$ group manifold [37].)

We note that if the fundamental charge is assumed to be $p$ rather than 1 then there is again a 1-form symmetry and a mixed anomaly involving charge conjugation and the center symmetry analogous to the case with $N = 1$.

# 3  3d Abelian Higgs Models

The Lagrangian is the same as in two dimensions (5) except there is no 3D theta angle. We first study models where the fundamental charge $p = 1$.

**Symmetries**: The central new ingredient in three dimensions–which in many ways replaces the theta angle–is the existence of a monopole charge. It has the covariant current

---

[10]Note that continuous symmetries cannot be broken in two dimensions and hence we do not include the option of spontaneous breaking of $PSU(N)$.

$j = da$, which is conserved simply because $d^2 = 0$. The charge measures the magnetic flux through space: $Q = \frac{1}{2\pi} \int_2 da$. Since charge conjugation maps $a \mapsto -a$, it also reverses $Q$,

$$CQC = -Q \, .$$

Therefore the total symmetry group, combining charge conjugation, flavor symmetry, and monopole charge is

$$C \ltimes (PSU(N) \times U(1)_T) \, .$$

The T stands for topological, since the monopole charge is a topological invariant of the gauge bundle, its Chern number.

**Phases**: We can deform the model by a mass term $m^2 \sum |\phi_i|^2$ which preserves all the symmetries of the problem and we can ask about the properties of the ground state in $\mathbb{R}^{2,1}$ as a function of $m^2$. If the mass squared is large and positive then we have a photon in 2+1 dimensions, which is equivalent to a compact scalar and $U(1)_T$ is spontaneously broken, so we have an $S^1$ of degenerate vacua. If the mass squared is large and negative then the gauge symmetry is Higgsed and we have a nonlinear sigma model with $\mathbb{CP}^{N-1}$ target space.

Note that charge conjugation is preserved in both phases but we see that either $U(1)_T$ or $PSU(N)$ are spontaneously broken. The model therefore does not have a disordered phase, at least not in the semi-classical limit. This is consistent with an anomaly, which we will soon discuss. The transition between the phase with broken $U(1)$ and broken $PSU(N)$ may be of first order or second order. For $N = 2$ there is a strong indication (see [11] and references therein) that it is second order.

An interesting question is whether we can break the symmetries further and still maintain the order in all the phases of the theory. Let us begin by trying to break $PSU(N)$ symmetry. We do this by adding mass deformations

$$\delta \mathcal{L} = \sum_j m_j^2 |\phi_j|^2. \tag{37}$$

For generic $m_j$, this explicitly breaks the $PSU(N)$ symmetry down to its diagonal subgroup. In general, it breaks it to a block diagonal subgroup. If one of the blocks is shape $1 \times 1$, say $m_1$, then we can take $m_1^2 \ll 0$ and $m_{j\neq 1}^2 \gg 0$ to find a trivial groundstate with no moduli. However, if all of the blocks are at least $2 \times 2$, then there is no way to Higgs $a$ without having some moduli left over. We will see that in these cases the anomaly remains non-trivial when restricted to the flavor subgroup.

Now let us consider breaking the $U(1)_T$ symmetry. This means that we can add monopole operators to the Lagrangian

$$\delta \mathcal{L} = \sum c_n M_n + c.c.$$

with $M_n$ carrying charge $n$ under $U(1)_T$. In the phase with $m^2 \gg e^2$ we have a free photon in 2+1 dimensions, and the monopoles induce a potential

$$V(\varphi) = \sum c_n e^{in\varphi} + c.c.$$

for the dual scalar $\varphi$. Generically there would be one trivial ground state. Note however that if we preserve a $\mathbb{Z}_2 \subset U(1)_T$, namely, we allow for even monopoles, then a two-fold degeneracy would necessarily remain. Hence, the phase remains nontrivial even if we preserve just a $\mathbb{Z}_2 \subset U(1)_T$.

Let us comment that we expect that at sufficiently high finite temperature the model is disordered. Equivalently, if instead of studying the model on $\mathbb{R}^{2,1}$ we were to study it on $S^1 \times \mathbb{R}^{1,1}$, then we would expect that for sufficiently small $S^1$ the Hamiltonian on $S^1 \times \mathbb{R}$ has a unique ground state. This is in contrast to the fact that a trivial phase cannot exist for a

sufficiently large $S^1$, as we have already seen semi-classically, and, more generally, due to the anomalies that we will soon discuss. Later we will discuss a variation of this model where some nontrivial order remains even at arbitrary finite temperature.

So far charge conjugation has not played a significant role in our discussion. However, imagine that we put the system at finite temperature and turn on a chemical potential for the $U(1)_T$ symmetry. To analyze this, let $A$ be a $U(1)$ gauge field that couples to the current $da$ meaning we add to the action

$$\frac{1}{2\pi} \int_3 A \wedge da \ . \tag{38}$$

If we now assume that along the thermal circle we have $\int_{S^1} A = \mu$ then we have a 1+1-dimensional model at long distances with

$$\mu = \theta_{2d} \ . \tag{39}$$

Such models, as we saw in the previous section, carry 't Hooft charge conjugation anomalies at $\theta_{2d} = \pi$. So we need to include the charge conjugation gauge field in our discussion if we want to correctly reproduce the physics at finite temperatures with chemical potentials. Similarly, in order to understand the domain wall theory in the phase with spontaneously broken $\mathbb{Z}_2 \subset U(1)_T$ we need to include the charge conjugation gauge field.

**Anomalies**: We use here the technical machinery developed in studying the 2d models. See that section for the definition of various cohomological objects and extended discussions about quantization conditions.

In the presence of a nontrivial $PSU(N) \rtimes \mathbb{Z}_2^C$ bundle $B'$ the fluxes of $da$ are quantized in units of $2\pi/N$ such that

$$\oint_2 \frac{D_{u_1} a}{2\pi} + \frac{u_2(B')}{N} \in \mathbb{Z} \, ,$$

for all closed surfaces, just as in the 1+1D models. Therefore, the minimal coupling (38) (modified to use the covariant derivative) does not represent an integer $U(1)_T$ charge and so is not invariant under large $U(1)_T$ gauge transformations. All of the global symmetries are preserved however, so to compute the anomaly we need merely to take the (covariant) differential of the offending term. We find its gauge variation precisely cancels the boundary variation of the 4d topological term

$$S_4^{anom} = -2\pi i \int_4 \frac{D_{u_1} A}{2\pi} \frac{u_2(B')}{N} \, , \tag{40}$$

Note that charge conjugation acts on the $U(1)_T$ gauge field $A$ by $A \mapsto -A$, so that $Ada$ is invariant under global charge conjugations, hence the appearance of $D_{u_1} A$. This also ensures (40) is well-defined. The combined bulk-boundary theory with this topological term respects the full gauge symmetry.

One can perform a simple consistency check of (40): Consider (38) in the case that the space-time manifold is of the type $M_2 \times S^1$ and $\int_{S^1} A = \mu$, such that $\mu \simeq \mu + 2\pi$. Reducing along the circle we land on the two dimensional Abelian Higgs model with $N$ fields of charge 1 and

$$\mu = \theta_{2d} \ .$$

Suppose we consider the continuous process where $\mu$ changes by $2\pi$. This is implemented by putting unit flux $\int F_A = 2\pi$ along the torus spanned by the $S^1$ in space-time and an auxiliary $S^1$ in (40). Integrating over this auxiliary torus we pick up the term $\frac{2\pi i}{N} \int_2 u_2(B)$. This exactly coincides with what we expect from (11).

Now consider breaking $U(1)_T$ to $\mathbb{Z}_n$, eg. by adding charge $n$ monopole operators to the path integral. So long as $n$ is even, the anomaly still remains. For example, consider $n = 2$. If we write the corresponding $\mathbb{Z}_2$ gauge field $A_2$, it is related to the $U(1)_T$ gauge field by $\exp \pi i \int A_2 = \exp i \int A$, so $F_A$ gets replaced with $\pi dA_2$ and the anomaly (40) becomes

$$2\pi i \frac{1}{N} \int_4 \frac{dA_2}{2} u_2(B') = \pi i \int_4 A_2 u_3(B') + 2\pi i \int_3 \frac{1}{2N} A_2 u_2(B'). \tag{41}$$

The first term on the RHS is a non-trivial class in $H^4(B\mathbb{Z}_2 \times BPSU(N), U(1))$ when $N$ is even, so it cannot be written as a gauge-invariant boundary term, and the second is a 3D counterterm. Here, as in section 2, when we discuss $u_3$, we should really include charge conjugation symmetry and therefore consider $u_3(B')$ as in equation (15). In general, if we break $U(1)$ to $\mathbb{Z}_m$, there is an anomaly $2\pi i \frac{1}{m} A_m u_3$ of order $gcd(N, m)$, which is non-trivial unless $N$ and $m$ are coprime.

When we reduce on a circle with twist $\int_{S^1} A_2 = 1$, we get the 2D model (5) at $\theta = \pi$. The anomaly polynomial (41) becomes

$$\pi i \int_3 u_3(B') + \pi i \int_2 u_2(B')/N. \tag{42}$$

The second term is the 2D counterterm (12) we added to the Lagrangian to preserve charge conjugation. The first term is the $C$-pinned mixed anomaly (15) we derived above. This reduction on the circle with a twist $\int_{S^1} A_2 = 1$ is equivalent to studying the theory at finite temperature and a chemical potential for the topological symmetry. Therefore, the fact that the anomaly remains nonzero after the reduction implies that the theory is ordered at any temperature in the presence of such a chemical potential.

We can make some consistency checks and present a few simple applications.

- From the anomaly it follows that that $\mathbb{Z}_2$ and $PSU(N)$ can be preserved in the vacuum only if it is a conformal field theory or a nontrivial TQFT. Indeed, in our phases that appear for large positive and large negative $m^2$ either the $\mathbb{Z}_2$ or the $PSU(N)$ were broken. If the transition is first order then both are broken at the transition point. If the transition is second order, then we have a massless theory.

- If we break the $PSU(N)$ symmetry to a block diagonal subgroup by adding mass terms like (37), $u_2(B)$ remains non-trivial as long as $N$ times the generator of $\pi_1(PSU(N))$

$$\begin{pmatrix} e^{i\theta/N} & 0 & 0 & 0 & 0 \\ 0 & e^{i\theta/N} & 0 & 0 & 0 \\ 0 & 0 & \ddots & 0 & 0 \\ 0 & 0 & 0 & e^{i\theta/N} & 0 \\ 0 & 0 & 0 & 0 & e^{i\theta/N - i\theta} \end{pmatrix},$$

  where $\theta \in [0, 2\pi)$, may be unwound around each block. This happens iff each block is size at least $2 \times 2$. This is consistent with our observation that a single $1 \times 1$ block can be used to Higgs $a$ without any moduli left over.

- In the phase with broken $\mathbb{Z}_2$ charged particles are confined and there is a domain wall. This phase in fact shares many similarities with QCD-like theories [23]. From the inflow polynomial (41) we learn that the domain wall theory, which is a 1+1 dimensional theory, must be nontrivial and it carries the mixed $PSU(N) \rtimes C$ anomaly $\frac{1}{2} \int_3 u_3(B')$, see (15). Since $PSU(N)$ is a continuous group, it cannot be broken in 1+1 dimensions and so the domain wall is rather constrained [14].

- Similarly, we can study the theory on a circle with chemical potential for the $\mathbb{Z}_2$ symmetry. The theory at long distances (compared to the circle radius) in 1+1 dimensions carries the above anomaly and hence it can never be disordered.

- Enhanced Symmetry: In the case of $N = 2$, i.e. when the Higgs phase is the $\mathbb{CP}^1$ model, the transition is the Néel-VBS transition and it is believed to be second order, associated to some conformal field theory. The conformal theory therefore has at least $C \ltimes (SO(3) \times U(1)_T)$ symmetry. Writing $U(1)_T = SO(2)$, we see this can be naturally embedded into $SO(5)$ and thus we can ask whether the anomaly (40) lifts to an $SO(5)$ anomaly. The answer turns out positive (see Appendix F); it lifts to the

$$\pi i \int_4 w_4(SO(5))$$

  anomaly. Therefore the anomalies are consistent with the existence of an enhanced $SO(5)$ symmetry at the fixed point [11].

- Lattice Models: The anomaly in the form (40) has a simple interpretation once one notes that $F_A/2\pi$ is Poincaré dual to the worldlines of $a$ charges on the 2+1D boundary. Indeed, when a monopole braids a flux in the magnetic symmetry, its wavefunction's phase rotates by $2\pi$, so the flux is identified with a gauge charge. The term (40) then says that this worldline is the boundary of the 1+1D SPT with cocycle $u_2(B)/N$, so the electric charge carries the $SU(N)$ fundamental representation (perhaps tensor some $PSU(N)$ reps). This observation is equivalent to the anomaly (40) and can be made in the lattice model of the RVB state [10] ($N = 2$). Indeed, a charge for the emergent gauge field (equivalent to our $a$) is an unpaired fermion, an $SU(2)$ fundamental. This implies that the lattice model in [10] also carries the anomaly (40), providing some more evidence that the Abelian Higgs model with $N = 2$ is a good effective field theory.

We would like to close this discussion with a general remark. A crucial idea here was that compactifying the 2+1 dimensional theory on a circle with a chemical potential, one ends up with a $\theta$ term in $1 + 1$ dimensions (39). Therefore, the various anomalies we discussed in 1+1 dimensions had to be uplifted to 2+1 dimensions and we also discussed how to include charge conjugation in this picture (41). A similar line of reasoning holds more generally. For example, consider $SU(N)$ gauge theory in 4+1 dimensions. It has a $U(1)_T$ symmetry whose current is topologically conserved and it also has a center symmetry. Reducing the model on a circle with a chemical potential for $U(1)_T$ leads to a four-dimensional theta term. Therefore, the 4+1 dimensional model has to have an anomaly which is the uplift of the one in [3], i.e. it is a mixed 't Hooft anomaly involving $U(1)_T$ and the center $\mathbb{Z}_N$ symmetry. Time reversal symmetry can be included in the same fashion that we included charge conjugation above. Such anomalies in $4 + 1$ dimensional gauge theories could be interesting to study further.

## 3.1 The 2+1d Abelian Higgs Model with $p > 1$

We now consider the abelian Higgs model with $N$ scalar fields with fundamental charge $p > 1$ coupled to a $U(1)$ gauge field $a$. On $\mathbb{R}^{2,1}$ the dynamics of the model is very similar to the case with $p = 1$. There is a Higgs phase and a confined phase and the transition between them may be continuous. The main new ingredient is the 1-form $\mathbb{Z}_p$ center symmetry. This symmetry shares an anomaly with the magnetic $U(1)_T$ symmetry which constrains the phase diagram and has important consequences for the behaviour of the theory at finite temperatures. We will see that this anomaly implies a phase diagram without any disordered phases even at finite temperature. Some kind of order persists over both quantum and thermal fluctuations.

For simplicity, let us discuss the new features of the center symmetry in the case of a single flavor $N = 1$. The Lagrangian reads

$$\frac{1}{4e^2}|da|^2 + |(d + ipa)\phi|^2 + m^2|\phi|^2 + \lambda|\phi|^4 .$$

It has three apparent symmetries

1. Magnetic symmetry $U(1)_T$, acting on the monopole operators but not the fields, with conserved current $da/2\pi$.

2. Charge conjugation $C$ acting on the fields by $\phi \mapsto \phi^*$, $a \mapsto -a$.

3. The 1-form center symmetry $B\mathbb{Z}_p$, acting on the fields by $a \to a + \lambda$, $\phi \mapsto e^{-is}\phi$, where $\lambda$ is a $U(1)$ connection and $s$ is an $\mathbb{R}/2\pi\mathbb{Z}$-valued scalar satisfying $ds = p\lambda$[11].

Charge conjugation anti-commutes with both symmetries, so the total symmetry algebra is $C \ltimes (B\mathbb{Z}_p \times U(1)_T)$.

This theory has two phases on $\mathbb{R}^{2,1}$:

1. Higgs phase for $m^2 \ll 0$: When $\phi$ attains a vev, the $U(1)$ gauge symmetry is broken to its $\mathbb{Z}_p$ subgroup which leaves $\phi$ untransformed. The theory is gapped but degenerate with a non-trivial topological field theory of a deconfined $\mathbb{Z}_p$ gauge field in the IR. The ground states of this gauge field are permuted by the center symmetry, so this is also spontaneously broken in this phase.

2. Coulomb phase for $m^2 \gg 0$: When the mass of $\phi$ is positive, we can integrate it out. This leaves us with just $a$, which can be dualized to a massless $U(1)$ scalar, whose $U(1)_T$ shift symmetry is spontaneously broken (and the scalar is the Goldstone mode).

To derive the anomaly, we couple to a background $U_T(1)$ field $A$:

$$\frac{1}{4e^2}|da|^2 + |D_a\phi|^2 + m^2|\phi|^2 + \lambda|\phi|^4 + A \wedge \frac{da}{2\pi}.$$

We also couple to a background $B\mathbb{Z}_p$ gauge field, which we write as a $\mathbb{Z}_p$ 2-form $K$. It twists the quantization rule for $a$ so now

$$\int_2 \frac{da}{2\pi} + \frac{K}{p} \in \mathbb{Z} \tag{43}$$

for all closed surfaces. This means that the minimal coupling to $A$ is now ill-quantized and transforms under a large $U(1)_T$ gauge transformation of $A$ parametrized by a $U(1)$ scalar $g$

$$\int_3 dg \frac{K}{p}, \tag{44}$$

which can contribute an arbitrary $p$th root of unity to the path integral weight, so we have an anomaly.

The physical picture here is that introducing monopole operators with nontrivial flux $\int F_A \neq 0$ breaks the $B\mathbb{Z}_p$ symmetry. This is because the source of the $B\mathbb{Z}_p$ symmetry is the fact that we only have charge $p$ particles, while the coupling term above shows that monopole operators ($\int dA \neq 0$) have charge 1.

---

[11]The pair $(\lambda, s)$, up to gauge transformations, is equivalent to a $\mathbb{Z}_p$ connection, see Appendix C.

While there is no 3D counterterm which can cancel the anomalous variation (44), it is canceled by the boundary variation of a 4D topological term

$$\frac{1}{p}\int_4 c(A)K, \tag{45}$$

where $c(A)$ is the Chern class of the $U(1)_T$ gauge bundle extended to a four manifold bounding our 3D spacetime. Compare with Section 2.1. This is a ($C$-invariant) non-trivial class in $H^4(B[U(1)_T \times B\mathbb{Z}_p], U(1))$.

The presence of this anomaly term in $\mathbb{R}^{2,1}$ explains why the phases we could see in our semi-classical analysis were all ordered. But now since $K$ is a two-form gauge field coupling to a 1-form global symmetry, upon a reduction on a circle, the center symmetry splits into a 1-form and a 0-form part on $\mathbb{R}^{1,1}$. The anomaly is now shared between $U(1)_T$ and the 0-form part:

$$\frac{1}{p}\int_3 c(A)B, \tag{46}$$

where $B = \int_{S^1} K$ is the background $\mathbb{Z}_p$ (1-form) gauge field for the induced 0-form center symmetry. This is a non-trivial class in $H^3(B[U(1)_T \times \mathbb{Z}_p], U(1))$.

In particular, the $1+1$ dimensional theory at distances much longer than the radius of the $S^1$ has a mixed anomaly between the $U(1)_T$ symmetry and the $\mathbb{Z}_p$ 0-form center symmetry. Therefore, for every radius of the $S^1$, the two-dimensional theory cannot be in a disordered phase. Hence, the theory on $S^1 \times \mathbb{R}^{1,1}$ is *always* ordered. This is reminiscent of Yang-Mills theory on a circle at $\theta = \pi$. The theory is always ordered. Here we see that similar phenomena can take place in slight modifications of the Abelian Higgs model in 3 dimensions.

## Acknowledgments

We would like to thank J. Cardy, J. Gomis, M. Metlitski, S. Sachdev, N. Seiberg, T. Sulejman-pasic, M. Unsal, and A. Zamolodchikov for useful discussions. Z.K. and A.S. are supported in part by an Israel Science Foundation center for excellence grant and by the I-CORE program of the Planning and Budgeting Committee and the Israel Science Foundation (grant number 1937/12). Z.K. is also supported by the ERC STG grant 335182 and by the Simons Foundation grant 488657 (Simons Collaboration on the Non-Perturbative Bootstrap). X.Z. is supported in part by NSF Grant No. PHY-1620628. R.T. is supported by an NSF GRFP grant and is grateful for the hospitality of the Weizmann Institute of Science, where this work began.

## A  Coupling Scalar $QED_2$ to a Two-Form Gauge Field

Consider the theory 2-dimensional theory:

$$\mathcal{L} = \frac{1}{4e^2}f \wedge \star f + \frac{i\theta}{2\pi}f,$$

where $f = da$ for $a$ a $U(1)$ gauge field. The theory has a 1-form global symmetry, with closed current $\star f$. Consider coupling the symmetry to a background 2-form gauge field $K$. Gauge transformations act as $K \rightarrow K + d\lambda^{(1)}$, $a \rightarrow a + \lambda^{(1)}$, where $\lambda$ is an arbitrary $U(1)$ connection.

In order for the coupled theory to be gauge invariant, we introduce the coupling by transforming $da \rightarrow D_K a = da - K$ (this is analogous to the minimal coupling of a free boson

$d\phi \wedge \star d\phi$ to its 0-form $U(1)$ symmetry by transforming $d\phi \rightarrow D_A\phi = d\phi - A$). The resulting Lagrangian is

$$\mathcal{L} = \frac{1}{4e^2}(f - K) \wedge \star (f - K) + \frac{i\theta}{2\pi}(f - K),$$

and so the $K$-dependent parts are

$$\mathcal{L}_K = \frac{1}{4e^2}K \wedge \star K - \frac{1}{2e^2}K \wedge \star f - \frac{i\theta}{2\pi}K.$$

The first two terms are $C$-even, but the last term is not $C$-invariant at $\theta = \pi$, transforming by $\int_2 iK$, which can contribute an arbitrary phase. To deal with this, let's look back at the $\theta$ term in the original Lagrangian. Under a gauge transformation, it transforms by

$$\frac{i\theta}{2\pi}\int_2 d\lambda.$$

This is non-trivial when $\lambda$ has a non-trivial Chern class, so it has something to do with large gauge transformations of $K$, while small gauge transformations, where $\lambda$ is a global 1-form, do not cause any transformation.

One way to describe the $U(1)$ 2-form gauge field $K$ in a way that mathematically separates small and large gauge transformations is using the theory of differential cocycles [38]. In this language, which is more or less a generalization of the Villain formalism, $K$ is described as a triple $(h, c_3, \Omega_3) \in C^2(X, \mathbb{R}) \times C^3(X, \mathbb{Z}) \times C^3(X, \mathbb{R})$ satisfying the differential cocycle equations

- $dc_3 = 0$,

- $d\Omega_3 = 0$,

- $dh = \Omega_3 - 2\pi c_3$,

where $\Omega_3$ represents the curvature of $K$, $h$ encodes its holonomy on closed surfaces, and $c_3$ is the Dixmier-Douady-Chern class, generalizing the Chern class of a complex line bundle. A gauge transformation is parametrized by a pair $(f_1, n_2) \in C^1(X, \mathbb{R}) \times C^2(X, \mathbb{Z})$ which represent the small and large gauge transformations, respectively. Under these transformations, the differential cocycle transforms by

- $c_3 \mapsto c_3 + dn_2$,

- $\Omega_3 \mapsto \Omega_3$,

- $h \mapsto h + df_1 - 2\pi n_2$.

We see that the variation of the $\theta$ term may be written

$$\frac{i\theta}{2\pi}\int_2 n_2,$$

and that this is cancelled by a bulk term

$$-\frac{i\theta}{2\pi}\int_3 c_3(K).$$

This can be related to the boundary holonomy $i\theta \int_2 K$ using the differential cocycle equation:

$$-\frac{i\theta}{2\pi}\int_3 c_3(K) = -\frac{i\theta}{2\pi}\Big[\int_3 \Omega_3(K) - \int_2 h(K)\Big].$$

# B    Holonomy Effective Potential in Scalar $QED_2$ on $\mathbb{R}^1 \times \mathbb{S}^1$

Suppose the scalar $QED_2$ is put on a ring with radius $R$, the vacuum energy of heavy scalar field will be the leading contribution to the effective action of the gauge field holonomy.

Let the scalar field be periodic in $x \to x + 2\pi R$

$$\Phi(x,t) = \sum_{n=-\infty}^{+\infty} e^{\frac{inx}{R}} \Phi_n(t) . \tag{47}$$

Then the Lagrangian

$$L = \sum_{n=-\infty}^{+\infty} (\frac{n}{R} + \frac{q}{2\pi R})^2 |\Phi_n|^2 + M^2 |\Phi_n|^2 + |\dot{\Phi}_n|^2$$

is just an infinite collection of harmonic oscillators. The vacuum energy

$$E = \sum_{n=-\infty}^{+\infty} \sqrt{(\frac{n}{R} + \frac{q}{2\pi R})^2 + M^2}$$

is formally divergent. To make sense of it, we zeta regularize it into

$$E(s) = \sum_{n=-\infty}^{+\infty} \left( (\frac{n}{R} + \frac{q}{2\pi R})^2 + M^2 \right)^{-s/2} . \tag{48}$$

$E(s)$ can be evaluated using the formula (see Appendix B of [39])

$$\sum_{n=-\infty}^{+\infty} [(n+b)^2 + a^2]^{-\lambda} = \pi^{1/2} a^{1-2\lambda} \frac{\Gamma(\lambda - 1/2)}{\Gamma(\lambda)}$$
$$+ 4\sin(\pi\lambda) \int_a^{\infty} du (u^2 - a^2)^{-\lambda} \mathrm{Re}\left( e^{2\pi(u+ib)} - 1 \right)^{-1} \tag{49}$$

and continue to $s = -1$. The result is

$$RE = \pi^{1/2} (MR)^{1-2\lambda} \frac{\Gamma(\lambda - 1/2)}{\Gamma(\lambda)} \Big|_{\lambda \to -1/2}$$
$$- 4 \int_{MR}^{\infty} du (u^2 - (MR)^2)^{1/2} \mathrm{Re}\left( e^{2\pi u + iq} - 1 \right)^{-1} . \tag{50}$$

The first term is divergent. But it does not depend on the holonomy so we can ignore it. The second term can be evaluated by rescaling $u = MRx$

$$-4(MR)^2 \int_1^{\infty} dx (x^2 - 1)^{1/2} \mathrm{Re}\left( e^{2\pi MRx + iq} - 1 \right)^{-1} .$$

In the large mass limit, the term $-1$ can be ignored in the real part and it evaluates into a Bessel K function

$$-4(MR)^2 \frac{K_1(2\pi MR)}{2\pi MR} \cos q \approx -\frac{\sqrt{MR}}{\pi} e^{-2\pi MR} \cos q .$$

Therefore the leading contribution in the effective action of the holonomy is

$$V(q) = -\frac{1}{\pi} \sqrt{\frac{M}{R}} e^{-2\pi MR} \cos q + \dots . \tag{51}$$

## C   Dual Description of the Stückelberg Action

We only need to focus on the term $\frac{1}{2}(d\varphi + pa)^2$ in the Stückelberg action which involves $\varphi$. The path integral over $\varphi$

$$\int \mathcal{D}\varphi \; e^{\int -\frac{1}{2}(d\varphi + pa)^2}$$

can be equivalently written as

$$\int \mathcal{D}\varphi \, \mathcal{D}A \, \mathcal{D}\tilde{\varphi} \; e^{\int -\frac{1}{2}(d\varphi + pa + A)^2 - \frac{i}{2\pi}\tilde{\varphi}dA} ,$$

where $A$ is an auxiliary gauge field and $\tilde{\varphi}$ is a Lagrange multiplier. When we first integrate out $\tilde{\varphi}$ and then integrate over $A$ we get back the original integral up to some overall normalization. To find the dual description, we just need to integrate in the opposite order. We can first use the gauge symmetry to gauge fix $\varphi = 0$. Suppressing the Faddeev-Popov integral related to the $\varphi$ integral, the path integral becomes

$$\int \mathcal{D}A \, \mathcal{D}\tilde{\varphi} \; e^{\int -\frac{1}{2}(pa + A)^2 - \frac{i}{2\pi}\tilde{\varphi}dA} = \int \mathcal{D}A \, \mathcal{D}\tilde{\varphi} \; e^{\int -\frac{1}{2}(pa + A - \frac{i}{2\pi}\star d\tilde{\varphi})^2 - \frac{1}{8\pi^2}(d\tilde{\varphi})^2 - \frac{ip}{2\pi}\tilde{\varphi}da} .$$

We can then shift $A' = A + pa - \frac{i}{2\pi} \star d\tilde{\varphi}$ and integrate out $A'$ which only contributes to some overall factor. We are then left with an integral over $\tilde{\varphi}$

$$\int \mathcal{D}\tilde{\varphi} \; e^{\int -\frac{1}{8\pi^2}(d\tilde{\varphi})^2 - \frac{ip}{2\pi}\tilde{\varphi}da}$$

and the dual Lagrangian of the Stückelberg model is therefore

$$\mathcal{L} = \frac{1}{4e^2}(da)^2 + \frac{1}{8\pi^2}(d\tilde{\varphi})^2 + \frac{ip}{2\pi}\tilde{\varphi}da . \tag{52}$$

Now let us consider what is the dual of the operator $e^{i\tilde{\varphi}}$ in the Stückelberg model. We proceed with the same method. The path integral of $\tilde{\varphi}$

$$\int \mathcal{D}\tilde{\varphi} \; e^{i\tilde{\varphi}(x)} e^{\int -\frac{1}{8\pi^2}(d\tilde{\varphi})^2 - \frac{ip}{2\pi}\tilde{\varphi}da}$$

can be written as

$$\int \mathcal{D}\tilde{\varphi} \, \mathcal{D}A \, \mathcal{D}\varphi \; e^{i\tilde{\varphi}(x)} e^{i\int_x A} e^{\int -\frac{1}{8\pi^2}(d\tilde{\varphi}+A)^2 + \frac{ip}{2\pi}(d\tilde{\varphi}+A)a + \frac{i}{2\pi}\varphi dA} ,$$

where $e^{i\int_x A}$ is a Wilson line starting at point $x$ to ensure the gauge invariance of $\tilde{\varphi}$ and $A$. We can use the gauge symmetry to set $\tilde{\varphi} = 0$. Meanwhile $\int_x A$ can be written as $\frac{1}{2\pi}\int \zeta A$ where $\zeta$ is a 1-form with $2\pi$ period such that

$$d\zeta = 2\pi\delta_x .$$

The integral now becomes

$$\int \mathcal{D}A \, \mathcal{D}\varphi \; e^{\int i\frac{1}{2\pi}\zeta A - \frac{1}{8\pi^2}A\star A + \frac{ip}{2\pi}Aa + \frac{i}{2\pi}\varphi dA} .$$

After completing the square and integrate out $A$, we obtain

$$\int \mathcal{D}\varphi \; e^{\int -\frac{1}{2}(-\zeta + ipa + d\varphi)^2} . \tag{53}$$

This looks identical to the Stückelberg action if we shift $d\varphi$ by a 1-form $\zeta$ but $\zeta$ actually has winding number 1. So the operator $e^{i\tilde{\varphi}}$ means for the Stückelberg model the prescription to remove a small disk around $x$ and twist $\varphi$ by $2\pi$ – it creates a vortex.

# D Circle Reduction of the Stückelberg Action

Consider the Stückelberg action:

$$\frac{1}{4e^2}(da)^2 + \frac{1}{8\pi^2}(\partial_\mu \tilde{\varphi})^2 + \frac{ip}{2\pi}\tilde{\varphi} da .$$

Using $F = da$ and completing the square, we obtain the action:

$$S = \int d^2x \left[ \frac{1}{2e^2}\left(F_{01} + \frac{ipe^2}{2\pi}\tilde{\varphi}\right)^2 + \frac{1}{8\pi^2}(\partial_\mu \tilde{\varphi})^2 + \frac{p^2e^2}{8\pi^2}\tilde{\varphi}^2 \right] .$$

We reduce on a circle of radius $R$. We choose Coulomb gauge, $\partial_1 A_1 = 0$, which makes $A_1$ spatially constant. Defining $A_1 = \frac{q(t)}{R}$, we obtain $\int A = q(t)$. The Gauss law is given by

$$\partial_1^2 A_0 = \frac{ipe^2}{2\pi}\partial_1 \tilde{\varphi} . \tag{54}$$

Expand the fields on the circle:

$$\tilde{\varphi} = \sum_{k \in \mathbb{Z}} \tilde{\varphi}_k(t) e^{i\frac{2\pi k}{R}x} ,$$

$$A_0 = \sum_{k \in \mathbb{Z}} A_{0,k}(t) e^{i\frac{2\pi k}{R}x} .$$

The Gauss law (54) then requires

$$A_{0,k} = \frac{pe^2}{2\pi k}\tilde{\varphi}_k, \quad k \neq 0 .$$

Plugging the expansions into the action we obtain:

$$S = \int dt \left[ \frac{1}{2e^2 R}\dot{q}^2 + \frac{R}{8\pi^2}(\dot{\tilde{\varphi}}_0)^2 + \frac{ip}{2\pi}\dot{q}\tilde{\varphi}_0 + \frac{R}{8\pi^2}\sum_{k \neq 0}\left((\dot{\tilde{\varphi}}_k)^2 + \left(\frac{4\pi^2 k^2}{R^2} + p^2 e^2\right)\tilde{\varphi}_k^2\right) \right] .$$

# E Dilute Gas Approximation

We start from the Lagrangian

$$\mathcal{L} = \frac{1}{4e^2}(da)^2 + \frac{1}{2}|(d + ipa)\phi|^2 + m^2|\phi|^2 + c_4|\phi|^4 .$$

When $m^2 < 0$, the theory is in the Higgs phase and $\phi$ can have a vev. Let us write

$$\phi = (v + \chi)e^{i\varphi} , \quad v = \sqrt{\frac{|m|^2}{2c_4}} ,$$

then the radial field $\chi$ and the gauge field $a$ gain masses from the Higgs mechanism,

$$m_\chi = 2|m| , \quad m_a = 2|m|\lambda^{-1} ,$$

where

$$\lambda = \frac{1}{p}\sqrt{\frac{4c_4}{e^2}} .$$

When $m \to \infty$ and keeping $m/\lambda$ fixed, the radial mode $\chi$ decouples and we are left with the Stückelberg model

$$\mathcal{L} = \frac{1}{4e^2}(da)^2 + \frac{v^2}{2}(d\varphi + pa)^2 \, .$$

We have chosen to set $v = 1$ in the main text.

When $m^2 < 0$ there also exist vortex solutions. For example, an one-vortex solution with flux $2\pi/p$ takes the form

$$\varphi = \theta \, , \quad a_i = \epsilon_{ij}A(r)x_j \, ,$$

with the boundary condition at $r \to 0$,

$$\chi \to 0 \, , \quad A \text{ finite}$$

and asymptotic behavior at $r \to \infty$,

$$\chi \to v \, , \quad A \to -\frac{1}{pr^2} \, .$$

The vortex has two characteristic lengths which are the inverse of $m_\chi$ and $m_a$. The two parameters respectively measure the distance it takes for $\chi$ and $A$ to reach their asymptotic values. There are three regimes one can consider depending on the different values of $\lambda$. When $\lambda > 1$, vortices experience repulsive forces among them and only those with one quanta of flux $\pm 2\pi/p$ are stable. This regime corresponds to the type-II superconductor. When $\lambda = 1$, there is no force. When $\lambda < 1$, the forces are attractive and all $n$-flux vortices are stable. This regime corresponds to the type-I superconductor.

Since we are considering the limit where the theory can be approximated by the Stückelberg model, both $m$ and $\lambda$ are taken large. This means we are in the repulsive regime and vortices have very small profiles. The repulsive potential energy among vortices at large separations (larger than the size of the hard core $m_a^{-1}$) can be computed classically (see e.g. [40])

$$E_{\text{repul}} = \frac{m_a^2 \pi}{p^2 e^2} \sum_{i \neq j} n_i n_j K_0(m_a |\vec{r}_i - \vec{r}_j|) \, , \tag{55}$$

where $n_i = \pm 1$ corresponding to vortex and anti-vortex. It should be pointed out that $K_0(m_a r)$ is the Green's function for a 2d Euclidean scalar field with mass $m_a$

$$(\nabla^2 - m_a^2)K_0(m_a r) = -2\pi \delta^{(2)}(\vec{r}) \, .$$

Now following Polyakov, we want to take a gas of such vortex-instantons and see its effect on correlators. When we perform the path integral, we should sum over all the configurations of vortex-instantons. This means the partition function takes the form

$$Z = \sum_{n,m} Z_{0,nm} \frac{e^{-(n+m)S_0}}{n!m!} \int \prod_{i=0}^{n} m_a^2 d^2 r_i^+ \int \prod_{j=0}^{m} m_a^2 d^2 r_j^- e^{-\frac{m_a^2 \pi}{p^2 e^2} \sum_{i<j}(K_0(m_a|\vec{r}_i^+ - \vec{r}_j^+|) + K_0(m_a|\vec{r}_i^- - \vec{r}_j^-|))}$$

$$\times e^{-\frac{m_a^2 \pi}{p^2 e^2} \sum_{i,j} K_0(m_a|\vec{r}_i^+ - \vec{r}_j^-|)} \, , \tag{56}$$

where $Z_{0,nm}$ is the partition function of the quantum fluctuation around the instanton background with $n$ vortices and $m$ anti-vortices and $S_0$ is the action needed to create a vortex or anti-vortex.

This partition function can be reproduced if we add to the dual Lagrangian of the Stückelberg model a term $m_a^2 e^{-S_0}(e^{i\tilde{\varphi}} + e^{-i\tilde{\varphi}})$ which has the interpretation of vortex and anti-vortex

creation operators (see Appendix C). Note when we keep $v$ explicit, the dual Lagrangian of Stückelberg model is

$$\mathcal{L} = \frac{1}{4e^2}(da)^2 + \frac{1}{8\pi^2 v^2}(d\tilde{\varphi})^2 + \frac{ip}{2\pi}\tilde{\varphi}da.$$

After we integrate out the gauge field $a$, the fluctuation part of $\tilde{\varphi}$ gets a mass term with mass exactly $m_a$. The rest of the integral is

$$\int \mathcal{D}\tilde{\varphi}\, e^{-\int \frac{1}{8\pi^2 v^2}(d\tilde{\varphi})^2 - \int \frac{e^2 p^2}{4\pi^2}\tilde{\varphi}^2 + \int m_a^2 e^{-S_0}(e^{i\tilde{\varphi}} + e^{-i\tilde{\varphi}})}$$

$$= \sum_{n,m} \frac{e^{-(n+m)S_0}}{n!m!} \int \mathcal{D}\tilde{\varphi}\, e^{-\int \frac{1}{8\pi^2 v^2}(d\tilde{\varphi})^2 - \int \frac{e^2 p^2}{4\pi^2}\tilde{\varphi}^2} \left(\int m_a^2 d^2 r e^{i\tilde{\varphi}}\right)^m \left(\int m_a^2 d^2 r e^{-i\tilde{\varphi}}\right)^n$$

$$= \sum_{n,m} \int \prod_{i=0}^{n} m_a^2 d^2 r_i^+ \int \prod_{j=0}^{m} m_a^2 d^2 r_j^- \frac{e^{-(n+m)S_0}}{n!m!} \int \mathcal{D}\tilde{\varphi}\, e^{-\frac{1}{8\pi^2 v^2}(d\tilde{\varphi})^2 - \frac{e^2 p^2}{4\pi^2}\tilde{\varphi}^2} e^{\sum_{i=1}^{m} i\tilde{\varphi}(r_i^+) - \sum_{j=1}^{n} i\tilde{\varphi}(r_j^-)}.$$

After completing the square and integrating out $\tilde{\varphi}$, the propagators give us exactly (56). We thus conclude the instanton gas induces for $\tilde{\varphi}$ an effective potential

$$V(\tilde{\varphi}) = -m_a^2\, e^{-S_0}(e^{i\tilde{\varphi}} + e^{-i\tilde{\varphi}}) = -2m_a^2\, e^{-S_0}\cos(\tilde{\varphi})\,. \tag{57}$$

# F $SO(5)$ Anomaly Polynomial and Gauging Charge Conjugation

We consider the fourth Stiefel-Whitney class $\frac{1}{2}w_4 \in H^4(BSO(5), U(1))$. When the $SO(5)$ bundle is a sum of the $SO(2) = U(1)$ bundle $A$ and the $SO(3) = PSU(2)$ bundle $B$, by the Whitney formula,

$$\frac{1}{2}w_4(A \oplus B) = \frac{1}{2}w_2(A)w_2(B) = \frac{1}{2}\frac{F_A}{2\pi}u_2(B),$$

in agreement with our anomaly (40) of the 3D $\mathbb{CP}^1$ model.

When we turn on a background $C$ gauge field $\mathfrak{a}$, the bundles $A$ and $B$ are promoted to an $O(2)$ bundle $\hat{A}$ and an $O(3)$ bundle $B'$ satisfying

$$w_1(\hat{A}) = w_1(B') = \mathfrak{a}\,.$$

It follows that $\hat{A} \oplus B'$ is an $SO(5)$ bundle and we can compute

$$\frac{1}{2}w_4(\hat{A} \oplus B') = \frac{1}{2}w_2(\hat{A})w_2(B') + \frac{1}{2}w_1(\hat{A})w_3(B')\,.$$

Using $w_1(\hat{A}) = w_1(B')$ and $w_1 w_3 = \frac{1}{2}dw_3$, the second term can be written as a boundary term:

$$\frac{1}{2}w_4(\hat{A} \oplus B') = \frac{1}{2}w_2(\hat{A})w_2(B') + \frac{1}{4}dw_3(B')\,.$$

When we restrict $A$ to the subgroup $\mathbb{Z}_2$, this embeds into $O(2)$ as

$$R = \begin{pmatrix} -1 & 0 \\ 0 & -1 \end{pmatrix},$$

while $C$ may be written

$$C = \begin{pmatrix} 1 & 0 \\ 0 & -1 \end{pmatrix}.$$

The total $O(2)$ bundle has connection

$$R^{A_2}C^{\mathfrak{a}} = \begin{pmatrix} (-1)^{A_2} & 0 \\ 0 & (-1)^{A_2+\mathfrak{a}} \end{pmatrix}.$$

In other words, the $O(2)$ bundle $\hat{A}$ is a direct sum of the $O(1)$ bundles $A_2$ and $A_2 + \mathfrak{a}$. We can then compute $w_2(\hat{A})$ in terms of $A$ and $\mathfrak{a}$ using the Whitney formula:

$$w_2(\hat{A}) = w_1(A_2)w_1(A_2 + \mathfrak{a}) = A_2^2 + A_2\mathfrak{a}.$$

Then we have

$$\frac{1}{2}w_4(\hat{A} \oplus B') = \frac{1}{2}(A_2^2 + A_2\mathfrak{a})w_2(B') + d(...).$$

Now, using $A_2^2 = \frac{dA_2}{2}$, $\mathfrak{a} = w_1(B')$, $\frac{dw_2}{2} = w_3 + w_1w_2$, and integrating by parts, we find

$$\frac{1}{2}w_4(\hat{A} \oplus B') = \frac{1}{2}A_2 w_3(B') + \frac{1}{4}d\big[(A_2 + \mathfrak{a})w_2(B')\big].$$

Upon compactifying along a circle with $\int_{S^1} A_2 = 1$, we get an $O(3) = SO(3) \rtimes C$ anomaly

$$\frac{1}{2}w_3(B') + \frac{1}{4}dw_2(B'),$$

in agreement with what we have derived in (42). These two observations prove that $w_4$ is the proper anomaly class. We note that if the symmetry is broken to any $SO(4)$ subgroup of $SO(5)$, then the anomaly $\frac{1}{2}w_4$ can be cured by a 3D counterterm.

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
