# Peer review of "Comments on Abelian Higgs Models and Persistent Order"

_SciPost Physics, doi:SciPost Phys. 6, 003 (2019)_

## Round 3 · Referee Report · Anonymous (Referee 1) · 2018-6-2

Strengths

1. This paper gives interesting demonstration of the fact that even simple bosonic systems can have anomalies of descrete symmetries. The anomalies have important implications about whether the symmetries can be disordered or not.

2.The models they discuss also are relevant to condensed matter systems.

Weaknesses

1. Some discussions are not clear.

Report

This paper demonstrates the existence of anomalies in abelian Higgs models in 2 and 3 dimensions. Although the systems are purely bosonic, there are anomalies of symmetries involving charge conjugation, flavor group, and 1-form center symmetry. The existence of anomalies implies that the theory cannot be completely disordered with a trivial gapped phase without topological order. The authors discuss the phases of the systems explicitly in weak coupling regions and some suggestions are made about strong coupling regions.

This is a very nice paper. Before recommending publication,
I would like to resuest some improvements for further clarity.

1.
In the introduction, it is claimed that sometimes it is possible to show that the phase transition must be second order. I assume that the authors are talking about the critical point $m^2_*$ of Figure 2. Although it is very plausible that the phase transition is second order there, I don't think it is proved by the arguments in the manuscript. The reason is that I don't see how to exclude the possibility that first order phase transition happens at any $\theta$. (Notice that first order transition can happen without any reason regardless of symmetries.) If that happens, there is an additional line in Figure 2 at $m^2 \sim 0$ with arbitrary $\theta$ which passes through $m^2_*$. This scenario is not likely, but unless the authors give a proof which excludes this possiblility, the claimed second order transition is a very plausible but not proved statement.

If it is possible to give a rigorous proof that the transition is second order in any strongly coupled QFT system, that would be extremely interesting. If the authors cannot give a rigorous proof, perhaps some of the statements in the introduction should be modified.

2.
I don't understand the relation between eq.(3) and (4). It seems that O(3) has nothing to do with the $Z_2 \subset SO(2)$ symmetry from the topological symmetry. If so, $w_3(O(3))$ does not contain any factor of $A$. However, the number of $A$ is increased from (3) to (4).
Why does this happen?

3.
The entire discussion between eq.(13) and the beginning of section 2.1 is very difficult to follow, and I would like to ask the authors to make the discussion clearer and accessible as far as possible. The points which are unclear include, but not restricted to, the following:

(i)A covariant derivative is introduced in an unlabeled equation to take care of the charge conjugation. But the charge conjugation is a discrete symmetry and hence its bundle is flat. The role of the covariant derivative should be explained.

(ii)In the paragraph below (15), $w_3(B')$ is explained by using the ``hedgehog number" of the adjoint bundle. However, at least for $O(3)$, the adjoint bundle is orientable since $O(3)=SO(3) \times Z_2$ and the $Z_2$ part acts trivially on the adjoint representation. Then, the hodgehog number is the Euler number of the orientable bundle, and this is zero in odd dimensions. I believe that $w_3(B')$ should be defined as the hodgehog number of not the adjoint rep., but the defining rep. of $O(3)$. If so, the explanation about the case of general $PSU(N)$ must be modified.

(iii)I don't understand some equations like $dw_2=w_3$, because $w_2$ is closed. More explanations are necessary.

(iv)The discussion about the case of odd $N$ is totally unclear.

(v)It is claimed above eq.(17) that $u_3(B)$ is an integral class. At least when $N=2$ and the global symmetry is $O(3)$, I don't know any such integral class of $O(3)$ bundle in three dimensions.

4.
If $c_3(K)$ of eq.(21) is $dK$, it is better to mention that around that equation.

5.
In eq.(29), the argument of cosine includes $\theta$. How does this happen? Very naively, in eq.(28), the coefficient of $da$ is $\tilde{\varphi}$ so it might seem that the instanton amplitude is multiplied by just $\exp(i \tilde{\varphi}}$ which does not include $\theta$. Of course this argument is too naive since the instanton cannot be described by the low energy effective theory (28). I believe the equation (28) is correct, but more explanation would be helpful about how $\theta$ appears.

6.
The authors regard characteristic polynomials in one-higher dimensions (i.e. bulk) as counterters of anomalies of the boundary theory. A better viewpoint is that if we extend $d$-dimensional manifold to $d+1$-dimensional manifold, the partition function is completely fixed. This philosophy was emphasized e.g. in
https://arxiv.org/abs/1508.04715
Then, it would be better to describe a completely well-defined partition function by extending the manifold to $d+1$-dimensions by combining bulk and boundary terms. The reason that I'm saying it is that equations like e.g. (45) alone are not well-defined if $K$ is defined only as a cohomology element.

For example, if charge conjugation is neglected, I believe that more precise description is as follows. We can define completely gauge invariant expression of
$\int_3 A \wedge da/2\pi $
as
$\int_4 dA \wedge da/2pi$
by extending 3-manifold to 4-manifold. It depends on how
the extention is done. The ambiguity is described by the above integral on closed manifolds. On closed manifolds, it is possible to replace da by $K$ as
$\int_4 dA \wedge da/2pi = -(1/p) \int_4 dA \wedge K$
This is the anomaly polynomial. Of course, an expression like (45) on manifolds with boundary is OK if we take and explicit representative of $K$ (i.e. not just as cohomology element) and impose some gauge invariance, and that is what the authors are doing. But then, it is nesessary to explain the precise framework to treat $K$ rather than describing it as a cohomology element.

Requested changes

I would like to ask the authors to address the points raised above to the extent that they agree with what I described. I leave it to the authors how much they modify the manuscript.

  • validity: top
  • significance: top
  • originality: top
  • clarity: good
  • formatting: perfect
  • grammar: perfect

Author:  Ryan Thorngren  on 2018-08-22  [id 308]

(in reply to Report 1 on 2018-06-02)

We thank the reviewer for their close reading of the paper and insightful remarks. We have edited the paper to address some of their concerns. For the benefit of the reader, we include here our response to the reviewer's comments.

  1. This is a fair point. So long as we cannot rule out nontrivial behavior at nonzero theta angle, there is no way to say for sure that something more complicated than a second order phase transition does not occur between the two weakly coupled regimes.

  2. Indeed, $O(3)$ and $SO(2)$ come as a direct product. The number of $A$ appears to increase after integration by parts because of an identity $A^2 = dA/2$, which holds for $\mathbb{Z}_2$ cocycles. Indeed, note that $A^2$ is actually a linear function of $A$ modulo 2, so the order in $A$ has not really increased.

3.i) Indeed, the connection is flat, but it does not mean that the covariant derivative is trivial. For instance, it is common to use cohomology with coefficients twisted by the orientation bundle $w_1$. This is a $\mathbb{Z}_2$ bundle, so any connection on it is flat (and all such connections are gauge-equivalent), however on a nonorientable $n$-manifold $X$ (ie. when such connections are flat but always nontrivial), $\mathbb{Z}$-valued cohomology computed with $D_{w_1}$ is not isomorphic to the usual $\mathbb{Z}$-valued cohomology computed with $d$. In particular, $H^n(X,\mathbb{Z}) = 0$ by virtue of nonorientability, but $H^n(X,\mathbb{Z}^{w_1}) = \mathbb{Z}$, generated by the twisted volume form.

3.ii) Let $C$ be the orientation-reversing element of $O(3)$. If $R_\theta \in SO(3)$ is a rotation about some axis by angle $\theta$, then $C R_\theta C = R_{-\theta}$. In terms of the Lie algebra, we can write $R_\theta = \exp \theta M$, where $M \in so(3)$. Thus the adjoint action of $C$ on the Lie algebra takes $M \mapsto -M$. This is because $O(3) \neq SO(3) \times \mathbb{Z}_2$, but rather $SO(3) \rtimes \mathbb{Z}_2$. Since $\det C = -1$, this means that the $O(3)$ adjoint bundle may be nonorientable.

Note also that the Euler class in odd dimensions may be nontrivial in 3 dimensions. For instance in the paper we mention the example of $\mathbb{RP}^3$ with the $O(3)$ bundle associated to $L \oplus L \oplus L$, where $L$ is the unique nontrivial real line bundle on $\mathbb{RP}^3$. One can compute by repeated application of the Whitney formula that $w_3(L \oplus L \oplus L) = w_1(L) w_2(L \oplus L) = w_1(L)^3$ generates $H^3(\mathbb{RP}^3,\mathbb{Z}) = \mathbb{Z}$.

3.iii) I agree these equations look strange. On the other hand, for certain mod N cohomology classes $[\alpha]$, we can only guarantee a representative cocycle with $d\alpha = 0$ mod $N$. In other words, $[d\alpha/N]$ may be a nontrivial (although always $N$-torsion) integer class.

3.iv) We regret that our reasoning here is hard to follow, but at this time we cannot see a better way of expressing it.

3.v) See 3.ii)

  1. Thank you, up to torsion this is indeed true.

  2. Equation (28) describes the theory in the deep Higgs phase and it does not know about theta. The leading theta dependence is in (29) and it is computed in appendix E.

  3. We agree this is a more precise characterization of anomaly in-flow. In interest of keeping our paper accessible we chose to express a simplified view of the anomalies. For those encountered in our study, this was totally adequate, especially since the anomalies could already be seen almost at the level of the action principle, without needing to compute the partition functions. We trust that the reader well-versed in anomalies can translate our story into their preferred way of thinking about things.

---

## Editorial Decision

published